# Inhomogeneous Hypergraph Clustering with Applications

**Pan Li**
Department ECE
UIUC
panli2@illinois.edu

**Olgica Milenkovic**
Department ECE
UIUC
milenkov@illinois.edu

## Abstract

Hypergraph partitioning is an important problem in machine learning, computer vision and network analytics. A widely used method for hypergraph partitioning relies on minimizing a normalized sum of the costs of partitioning hyperedges across clusters. Algorithmic solutions based on this approach assume that different partitions of a hyperedge incur the same cost. However, this assumption fails to leverage the fact that different subsets of vertices within the same hyperedge may have different structural importance. We hence propose a new hypergraph clustering technique, termed inhomogeneous hypergraph partitioning, which assigns different costs to different hyperedge cuts. We prove that inhomogeneous partitioning produces a quadratic approximation to the optimal solution if the inhomogeneous costs satisfy submodularity constraints. Moreover, we demonstrate that inhomogenous partitioning offers significant performance improvements in applications such as structure learning of rankings, subspace segmentation and motif clustering.

## 1   Introduction

Graph partitioning or clustering is a ubiquitous learning task that has found many applications in statistics, data mining, social science and signal processing [1, 2]. In most settings, clustering is formally cast as an optimization problem that involves entities with different pairwise similarities and aims to maximize the total "similarity" of elements within clusters [3, 4, 5], or simultaneously maximize the total similarity within cluster and dissimilarity between clusters [6, 7, 8]. Graph partitioning may be performed in an agnostic setting, where part of the optimization problem is to automatically learn the number of clusters [6, 7].

Although similarity among entities in a class may be captured via pairwise relations, in many real-world problems it is necessary to capture joint, higher-order relations between subsets of objects. From a graph-theoretic point of view, these higher-order relations may be described via hypergraphs, where objects correspond to vertices and higher-order relations among objects correspond to hyperedges. The vertex clustering problem aims to minimize the similarity across clusters and is referred to as hypergraph partitioning. Hypergraph clustering has found a wide range of applications in network motif clustering, semi-supervised learning, subspace clustering and image segmentation. [8, 9, 10, 11, 12, 13, 14, 15].

Classical hypergraph partitioning approaches share the same setup: A nonnegative weight is assigned to every hyperedge and if the vertices in the hyperedge are placed across clusters, a cost proportional to the weight is charged to the objective function [9, 11]. We refer to this clustering procedure as *homogenous hyperedge clustering* and refer to the corresponding partition as a *homogeneous partition (H-partition)*. Clearly, this type of approach prohibits the use of information regarding how different vertices or subsets of vertices belonging to a hyperedge contribute to the higher-order relation. A more appropriate formulation entails charging different costs to different cuts of the

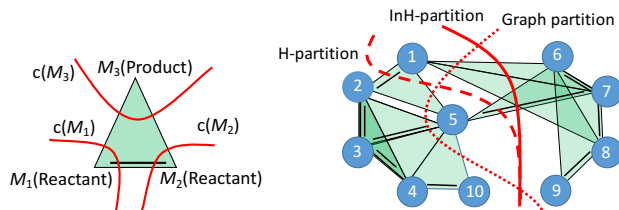

Figure 1: Clusters obtained using homogenous and inhomogeneous hypergraph partitioning and graph partitioning (based on pairwise relations). Left: Each reaction is represented by a hyperedge. Three different cuts of a hyperedge are denoted by $c(M_3), c(M_1)$, and $c(M_2)$, based on which vertex is "isolated" by the cut. The graph partition only takes into account pairwise relations between reactants, corresponding to $w(c(M_3)) = 0$. The homogenous partition enforces the three cuts to have the same weight, $w(c(M_3)) = w(c(M_1)) = w(c(M_2))$, while an inhomogenous partition is not required to satisfy this constraint. Right: Three different clustering results based on optimally normalized cuts for a graph partition, a homogenous partition (H-partition) and an inhomogenous partition (InH-partition) with $0.01\, w(c(M_1)) \le w(c(M_3)) \le 0.44\, w(c(M_1))$.

hyperedges, thereby endowing hyperedges with vector weights capturing these costs. To illustrate the point, consider the example of metabolic networks [16]. In these networks, vertices describe metabolites while edges describe transformative, catalytic or binding relations. Metabolic reactions are usually described via equations that involve more than two metabolites, such as $M_1 + M_2 \rightarrow M_3$. Here, both metabolites $M_1$ and $M_2$ need to be present in order to complete the reaction that leads to the creation of the product $M_3$. The three metabolites play different roles: $M_1, M_2$ are reactants, while $M_3$ is the product metabolite. A synthetic metabolic network involving reactions with three reagents as described above is depicted in Figure 1, along with three different partitions induced by a homogeneous, inhomogeneous and classical graph cut. As may be seen, the hypergraph cuts differ in terms of how they split or group pairs of reagents. The inhomogeneous clustering preserves all but one pairing, while the homogenous clustering splits two pairings. The graph partition captures only pairwise relations between reactants and hence, the optimal normalized cut over the graph splits six reaction triples. The differences between inhomogenous, homogenous, and pairwise-relation based cuts are even more evident for large graphs and they may lead to significantly different partitioning performance in a number of important partitioning applications.

The problem of inhomogeneous hypergraph clustering has not been previously studied in the literature. The main results of the paper are efficient algorithms for inhomogeneous hypergraph partitioning with theoretical performance guarantees and extensive testing of inhomogeneous partitioning in applications such as hierarchical biological network studies, structure learning of rankings and subspace clustering[1] (All proofs and discussions of some applications are relegated to the Supplementary Material). The algorithmic methods are based on transforming hypergraphs into graphs and subsequently performing spectral clustering based on the normalized Laplacian of the derived graph. A similar approach for homogenous clustering has been used under the name of Clique Expansion [14]. However, the projection procedure, which is the key step of Clique Expansion, differs significantly from the projection procedure used in our work, as the inhomogenous clustering algorithm allows non-uniform expansion of one hyperedge while Clique Expansion only allows for uniform expansions. A straightforward analysis reveals that the normalized hypergraph cut problem [11] and the normalized Laplacian homogeneous hypergraph clustering algorithms [9, 11] are special cases of our proposed algorithm, where the costs assigned to the hyperedges take a very special form. Furthermore, we show that when the costs of the proposed inhomogeneous hyperedge clustering are *submodular*, the projection procedure is guaranteed to find a constant-approximation solution for several graph-cut related entities. Hence, the inhomogeneous clustering procedure has the same quadratic approximation properties as spectral graph clustering [17].

## 2 Preliminaries and Problem Formulation

A hypergraph $\mathcal{H} = (V, E)$ is described in terms of a vertex set $V = \{v_1, v_2, ..., v_n\}$ and a set of hyperedges $E$. A hyperedge $e \in E$ is a subset of vertices in $V$. For an arbitrary set $S$, we let $|S|$ stand for the cardinality of the set, and use $\delta(e) = |e|$ to denote the size of a hyperedge. If for all $e \in E$, $\delta(e)$ equals a constant $\Delta$, the hypergraph is called a $\Delta$-uniform hypergraph.

Let $2^e$ denote the power set of $e$. An inhomogeneous hyperedge (InH-hyperedge) is a hyperedge with an associated weight function $w_e : 2^e \to \mathbb{R}_{\geq 0}$. The weight $w_e(S)$ indicates the cost of cutting/partitioning the hyperedge $e$ into two subsets, $\bar{S}$ and $e/S$. A consistent weight $w_e(S)$ satisfies the following properties: $w_e(\emptyset) = 0$ and $w_e(S) = w_e(e/S)$. The definition also allows $w_e(\cdot)$ to be enforced only for a subset of $2^e$. However, for singleton sets $S = \{v\} \in e$, $w_e(\{v\})$ has to be specified. The degree of a vertex $v$ is defined as $d_v = \sum_{e:\, v \in e} w_e(\{v\})$, while the volume of a subset of vertices $S \subseteq V$ is defined as

$$\mathrm{vol}_{\mathcal{H}}(S) = \sum_{v \in S} d_v. \tag{1}$$

Let $(S, \bar{S})$ be a partition of the vertices $V$. Define the hyperedge boundary of $S$ as $\partial S = \{e \in E | e \cap S \neq \emptyset, e \cap \bar{S} \neq \emptyset\}$ and the corresponding set volume as

$$\mathrm{vol}_{\mathcal{H}}(\partial S) = \sum_{e \in \partial S} w_e(e \cap S) = \sum_{e \in E} w_e(e \cap S), \tag{2}$$

where the second equality holds since $w_e(\emptyset) = w_e(e) = 0$. The task of interest is to minimize the *normalized cut* NCut of the hypergraph with InH-hyperedges, i.e., to solve the following optimization problem

$$\arg\min_S \mathrm{NCut}_{\mathcal{H}}(S) = \arg\min_S \mathrm{vol}_{\mathcal{H}}(\partial S) \left( \frac{1}{\mathrm{vol}_{\mathcal{H}}(S)} + \frac{1}{\mathrm{vol}_{\mathcal{H}}(\bar{S})} \right). \tag{3}$$

One may also extend the notion of InH hypergraph partitioning to $k$-way InH-partition. For this purpose, we let $(S_1, S_2, ..., S_k)$ be a $k$-way partition of the vertices $V$, and define the $k$-way normalized cut for inH-partition according to

$$\mathrm{NCut}_{\mathcal{H}}(S_1, S_2, ..., S_k) = \sum_{i=1}^{k} \frac{\mathrm{vol}_{\mathcal{H}}(\partial S_i)}{\mathrm{vol}_{\mathcal{H}}(S_i)}. \tag{4}$$

Similarly, the goal of a $k$-way inH-partition is to minimize $\mathrm{NCut}_{\mathcal{H}}(S_1, S_2, ..., S_k)$. Note that if $\delta(e) = 2$ for all $e \in E$, the above definitions are consistent with those used for graphs [18].

## 3 Inhomogeneous Hypergraph Clustering Algorithms

Motivated by the homogeneous clustering approach of [14], we propose an inhomogeneous clustering algorithm that uses three steps: 1) Projecting each InH-hyperedge onto a subgraph; 2) Merging the subgraphs into a graph; 3) Performing classical spectral clustering based on the normalized Laplacian (described in the Supplementary Material, along with the complexity of all algorithmic steps). The novelty of our approach is in introducing the inhomogenous clustering constraints via the projection step, and stating an optimization problem that provides the provably best weight splitting for projections. All our theoretical results are stated for the NCut problem, but the proposed methods may be used as heuristics for $k$-way NCuts.

Suppose that we are given a hypergraph with inhomogeneous hyperedge weights, $\mathcal{H} = (V, E, \mathbf{w})$. For each InH-hyperedge $(e, w_e)$, we aim to find a complete subgraph $G_e = (V^{(e)}, E^{(e)}, w^{(e)})$ that "best" represents this InH-hyperedge; here, $V^{(e)} = e$, $E^{(e)} = \{\{v, \tilde{v}\} | v, \tilde{v} \in e, v \neq \tilde{v}\}$, and $w^{(e)} : E^{(e)} \to R$ denotes the hyperedge weight vector. The goal is to find the graph edge weights that provide the best approximation to the split hyperedge weight according to:

$$\min_{w^{(e)}, \beta^{(e)}} \beta^{(e)} \text{ s.t. } w_e(S) \leq \sum_{v \in S, \tilde{v} \in e/S} w_{v\tilde{v}}^{(e)} \leq \beta^{(e)} \, w_e(S), \text{ for all } S \in 2^e \text{ s.t. } w_e(S) \text{ is defined.} \tag{5}$$

Upon solving for the weights $w^{(e)}$, we construct a graph $\mathcal{G} = (V, E_o, w)$, where $V$ are the vertices of the hypergraph, $E_o$ is the complete set of edges, and where the weights $w_{v\tilde{v}}$, are computed via

$$w_{v\tilde{v}} \triangleq \sum_{e \in E} w_{v\tilde{v}}^{(e)}, \quad \forall \{v, \tilde{v}\} \in E_o. \tag{6}$$

This step represents the projection weight merging procedure, which simply reduces to the sum of weights of all hyperedge projections on a pair of vertices. Due to the linearity of the volumes (1) and boundaries (2) of sets $S$ of vertices, for any $S \subset V$, we have

$$\text{Vol}_{\mathcal{H}}(\partial S) \leq \text{Vol}_{\mathcal{G}}(\partial S) \leq \beta^* \text{Vol}_{\mathcal{H}}(\partial S), \quad \text{Vol}_{\mathcal{H}}(S) \leq \text{Vol}_{\mathcal{G}}(S) \leq \beta^* \text{Vol}_{\mathcal{H}}(S), \quad (7)$$

where $\beta^* = \max_{e \in E} \beta^{(e)}$. Applying spectral clustering on $\mathcal{G} = (V, E_o, w)$ produces the desired partition $(S^*, \bar{S}^*)$. The next result is a consequence of combining the bounds of (7) with the approximation guarantees of spectral graph clustering (Theorem 1 [17]).

**Theorem 3.1.** If the optimization problem (5) is feasible for all InH-hyperedges and the weights $w_{v\tilde{v}}$ obtained from (6) are nonnegative for all $\{v, \tilde{v}\} \in E_o$, then $\alpha^* = \text{NCut}_{\mathcal{H}}(S^*)$ satisfies

$$(\beta^*)^3 \alpha_{\mathcal{H}} \geq \frac{(\alpha^*)^2}{8} \geq \frac{\alpha_{\mathcal{H}}^2}{8}. \quad (8)$$

where $\alpha_{\mathcal{H}}$ is the optimal value of normalized cut of the hypergraph $\mathcal{H}$.

There are no guarantees that the $w_{v\tilde{v}}$ will be nonnegative: The optimization problem (5) may result in solutions $w^{(e)}$ that are negative. The performance of spectral methods in the presence of negative edge weights is not well understood [19, 20]; hence, it would be desirable to have the weights $w_{v\tilde{v}}$ generated from (6) be nonnegative. Unfortunately, imposing nonngativity constraints in the optimization problem may render it infeasible. In practice, one may use $(w_{v\tilde{v}})_+ = \max\{w_{v\tilde{v}}, 0\}$ to remove negative weights (other choices, such as $(w_{v\tilde{v}})_+ = \sum_e (w_{v\tilde{v}}^{(e)})_+$ do not appear to perform well). This change invalidates the theoretical result of Theorem 3.1, but provides solutions with very good empirical performance. The issues discussed are illustrated by the next example.

*Example* 3.1. Let $e = \{1, 2, 3\}$, $(w_e(\{1\}), w_e(\{2\}), w_e(\{3\})) = (0, 0, 1)$. The solution to the weight optimization problem is $(\beta^{(e)}, w_{12}^{(e)}, w_{13}^{(e)}, w_{23}^{(e)}) = (1, -1/2, 1/2, 1/2)$. If all components $w^{(e)}$ are constrained to be nonnegative, the optimization problem is infeasible. Nevertheless, the above choice of weights is very unlikely to be encountered in practice, as $w_e(\{1\}), w_e(\{2\}) = 0$ indicates that vertices 1 and 2 have no relevant connections within the given hyperedge $e$, while $w_e(\{3\}) = 1$ indicates that vertex 3 is strongly connected to 1 and 2, which is a contradiction. Let us assume next that the negative weight is set to zero. Then, we adjust the weights $((w_{12}^{(e)})_+, w_{13}^{(e)}, w_{23}^{(e)}) = (0, 1/2, 1/2)$, which produce clusterings ((1,3)(2)) or ((2,3)(1)); both have zero costs based on $w_e$.

Another problem is that arbitrary choices for $w_e$ may cause the optimization problem to be infeasible (5) even if negative weights of $w^{(e)}$ are allowed, as illustrated by the following example.

*Example* 3.2. Let $e = \{1, 2, 3, 4\}$, with $w_e(\{1, 4\}) = w_e(\{2, 3\}) = 1$ and $w_e(S) = 0$ for all other choices of sets $S$. To force the weights to zero, we require $w_{v\tilde{v}}^{(e)} = 0$ for all pairs $v\tilde{v}$, which fails to work for $w_e(\{1, 4\}), w_e(\{2, 3\})$. For a hyperedge $e$, the degrees of freedom for $w_e$ are $2^{\delta(e)-1} - 1$, as two values of $w_e$ are fixed, while the other values are paired up by symmetry. When $\delta(e) > 3$, we have $\binom{\delta(e)}{2} < 2^{\delta(e)-1} - 1$, which indicates that the problem is overdetermined/infeasible.

In what follows, we provide sufficient conditions for the optimization problem to have a feasible solution with nonnegative values of the weights $w^{(e)}$. Also, we provide conditions for the weights $w_e$ that result in a small constant $\beta^*$ and hence allow for quadratic approximations of the optimum solution. Our results depend on the availability of information about the weights $w_e$: In practice, the weights have to be inferred from observable data, which may not suffice to determine more than the weight of singletons or pairs of elements.

**Only the values of $w_e(\{v\})$ are known.** In this setting, we are only given information about how much each node contributes to a higher-order relation, i.e., we are only given the values of $w_e(\{v\})$, $v \in V$. Hence, we have $\delta(e)$ costs (equations) and $\delta(e) \geq 3$ variables, which makes the problem underdetermined and easy to solve. The optimal $\beta^e = 1$ is attained by setting for all edges $\{v, \tilde{v}\}$

$$w_{v\tilde{v}}^{(e)} = \frac{1}{\delta(e) - 2} [w_e(\{v\}) + w_e(\{\tilde{v}\})] - \frac{1}{(\delta(e) - 1)(\delta(e) - 2)} \sum_{v' \in e} w_e(\{v'\}). \quad (9)$$

The components of $w_e(\cdot)$ with positive coefficients in (3) are precisely those associated with the endpoints of edges $v\tilde{v}$. Using simple algebraic manipulations, one can derive the conditions under which the values $w_{v\tilde{v}}^{(e)}$ are nonnegative, and these are presented in the Supplementary Material.

The solution to (9) produces a perfect projection with $\beta^{(e)} = 1$. Unfortunately, one cannot guarantee that the solution is nonnegative. Hence, the question of interest is to determine for what types of cuts can one can deviate from a perfect projection but ensure that the weights are nonnegative. The proposed approach is to set the unspecified values of $w_e(\cdot)$ so that the weight function becomes submodular, which guarantees nonnegative weights $w_{v\tilde{v}}^e$ that can constantly approximate $w_e(\cdot)$, although with a larger approximation constant $\beta$.

**Submodular weights** $w_e(S)$. As previously discussed, when $\delta(e) > 3$, the optimization problem (5) may not have any feasible solutions for arbitrary choices of weights. However, we show next that if the weights $w_e$ are *submodular*, then (5) always has a nonnegative solution. We start by recalling the definition of a submodular function.

**Definition 3.2.** A function $w_e : 2^e \to \mathbb{R}_{\geq 0}$ that satisfies

$$w_e(S_1) + w_e(S_2) \geq w_e(S_1 \cap S_2) + w_e(S_1 \cup S_2) \quad \text{for all } S_1, S_2 \in 2^e,$$

is termed submodular.

**Theorem 3.3.** If $w_e$ is submodular, then

$$w_{v\tilde{v}}^{*(e)} = \sum_{S \in 2^e/\{\emptyset, e\}} \left[ \frac{w_e(S)}{2|S|(\delta(e) - |S|)} 1_{|\{v,\tilde{v}\} \cap S| = 1} \right. \tag{10}$$

$$\left. - \frac{w_e(S)}{2(|S|+1)(\delta(e) - |S| - 1)} 1_{|\{v,\tilde{v}\} \cap S| = 0} - \frac{w_e(S)}{2(|S|-1)(\delta(e) - |S| + 1)} 1_{|\{v,\tilde{v}\} \cap S| = 2} \right]$$

is nonnegative. For $2 \leq \delta(e) \leq 7$, the function above is a feasible solution for the optimization problem (5) with parameters $\beta^{(e)}$ listed in Table 1.

Table 1: Feasible values of $\beta^{(e)}$ for $\delta^{(e)}$

| $|\delta(e)|$ | 2 | 3 | 4 | 5 | 6 | 7 |
|---|---|---|---|---|---|---|
| $\beta$ | 1 | 1 | 3/2 | 2 | 4 | 6 |

Theorem 3.3 also holds when some weights in the set $w_e$ are not specified, but may be completed to satisfy submodularity constraints (See Example 3.3).

*Example* 3.3. Let $e = \{1, 2, 3, 4\}$, $(w_e(\{1\}), w_e(\{2\}), w_e(\{3\}), w_e(\{4\})) = (1/3, 1/3, 1, 1)$. Solving (9) yields $w_{12}^{(e)} = -1/9$ and $\beta^{(e)} = 1$. By completing the missing components in $w_e$ as $(w_e(\{1,2\}), w_e(\{1,3\}), w_e(\{1,4\})) = (2/3, 1, 1)$ leads to submodular weights (Observe that completions are not necessarily unique). Then, the solution of (10) gives $w_{12}^{(e)} = 0$ and $\beta^{(e)} \in (1, 2/3]$, which is clearly larger than one.

*Remark* 3.1. It is worth pointing out that $\beta = 1$ when $\delta(e) = 3$, which asserts that homogeneous triangle clustering may be performed via spectral methods on graphs without any weight projection distortion [9]. The above results extend this finding to the inhomogeneous case whenever the weights are submodular. In addition, triangle clustering based on random walks [21] may be extended to the inhomogeneous case.

Also, (10) lead to an optimal approximation ratio $\beta^{(e)}$ if we restrict $w^{(e)}$ to be a linear mapping of $w_e$, which is formally stated next.

**Theorem 3.4.** Suppose that for all pairs of $\{v, \tilde{v}\} \in E_o$, $w_{v\tilde{v}}^{(e)}$ is a linear function of $w_e$, denoted by $w_{v\tilde{v}}^{(e)} = f_{v\tilde{v}}(w_e)$, where $\{f_{v\tilde{v}}\}_{\{v\tilde{v} \in E^{(e)}\}}$ depends on $\delta(e)$ but not on $w_e$. Then, when $\delta(e) \leq 7$, the optimal values of $\beta$ for the following optimization problem depend only on $\delta(e)$, and are equal to those listed in Table 1.

$$\min_{\{f_{v\tilde{v}}\}_{\{v,\tilde{v}\} \in E_o}, \beta} \quad \max_{\text{submodular } w_e} \quad \beta \tag{11}$$

$$\text{s.t.} \quad w_e(S) \leq \sum_{v \in S, \tilde{v} \in e/S} f_{v\tilde{v}}(w_e) \leq \beta w_e(S), \quad \text{for all } S \in 2^e.$$

*Remark* 3.2. Although we were able to prove feasibility (Theorem 3.3) and optimality of linear solutions (Theorem 3.4) only for small values of $\delta(e)$, we conjecture the results to be true for all $\delta(e)$.

The following theorem shows that if the weights $w_e$ of hyperedges in a hypergraph are generated from graph cuts of a latent weighted graph, then the projected weights of hyperedges are proportional to the corresponding weights in the latent graph.

**Theorem 3.5.** Suppose that $G_e = (V^{(e)}, E^{(e)}, w^{(e)})$ is a latent graph that generates hyperedge weights $w_e$ according to the following procedure: for any $S \subseteq e$, $w_e(S) = \sum_{v \in S, \tilde{v} \in e/S} w_{v\tilde{v}}^{(e)}$. Then, equation (10) establishes that $w_{v\tilde{v}}^{*(e)} = \beta^{(e)} w_{v\tilde{v}}^{(e)}$, for all $v\tilde{v} \in E^{(e)}$, with $\beta^{(e)} = \frac{2^{\delta(e)} - 2}{\delta(e)(\delta(e) - 1)}$.

Theorem 3.5 establishes consistency of the linear map (10), and also shows that the min-max optimal approximation ratio for linear functions equals $\Omega(2^{\delta(e)}/\delta(e)^2)$. An independent line of work [22], based on Gomory-Hu trees (non-linear), established that submodular functions represent nonnegative solutions of the optimization problem (5) with $\beta^{(e)} = \delta_e - 1$. Therefore, an unrestricted solution of the optimization problem (5) ensures that $\beta^{(e)} \leq \delta_e - 1$.

As practical applications almost exclusively involve hypergraphs with small, constant $\delta(e)$, the Gomory-Hu tree approach in this case is suboptimal in approximation ratio compared to (10). The expression (10) can be rewritten as $w^{*(e)} = M\,w_e$, where $M$ is a matrix that only depends on $\delta(e)$. Hence, the projected weights can be computed in a very efficient and simple manner, as opposed to constructing the Gomory-Hu tree or solving (5) directly. In the rare case that one has to deal with hyperedges for which $\delta(e)$ is large, the Gomory-Hu tree approach and a solution of (5) may be preferred.

## 4 Related Work and Discussion

One contribution of our work is to introduce the notion of an inhomogenous partition of hyperedges and a new hypergraph projection method that accompanies the procedure. Subsequent edge weight merging and spectral clustering are standardly used in hypergraph clustering algorithms, and in particular in Zhou's normalized hypergraph cut approach [11], Clique Expansion, Star Expansion and Clique Averaging [14]. The formulation closest to ours is Zhou's method [11]. In the aforementioned hypergraph clustering method for H-hyperedges, each hyperedge $e$ is assigned a scalar weight $w_e^{\mathrm{H}}$. For the projection step, Zhou used $w_e^{\mathrm{H}}/\delta(e)$ for the weight of each pair of endpoints of $e$. If we view the H-hyperedge as an InH-hyperedge with weight function $w_e$, where $w_e(S) = w_e^{\mathrm{H}}|S|(\delta(e) - |S|)/\delta(e)$ for all $S \in 2^e$, then our definition of the volume/cost of the boundary (2) is identical to that of Zhou's. With this choice of $w_e$, the optimization problem (5) outputs $w_{v\tilde{v}}^{(e)} = w_e^{\mathrm{H}}/\delta(e)$, with $\beta^{(e)} = 1$, which are the same values as those obtained via Zhou's projection. The degree of a vertex in [11] is defined as $d_v = \sum_{e \in E} h(e, v) w_e^{\mathrm{H}} = \sum_{e \in E} \frac{\delta(e)}{\delta(e) - 1} w_e(\{v\})$, which is a weighted sum of the $w_e(\{v\})$ and thus takes a slightly different form when compared to our definition. As a matter of fact, for uniform hypergraphs, the two forms are same. Some other hypergraph clustering algorithms, such as Clique expansion and Star expansion, as shown by Agarwal et al. [23], represent special cases of our method for uniform hypergraphs as well.

The Clique Averaging method differs substantially from all the aforedescribed methods. Instead of projecting each hyperedge onto a subgraph and then combining the subgraphs into a graph, the algorithm performs a one-shot projection of the whole hypergraph onto a graph. The projection is based on a $\ell_2$-minimization rule, which may not allow for constant-approximation solutions. It is unknown if the result of the procedure can provide a quadratic approximation for the optimum solution. Clique Averaging also has practical implementation problems and high computational complexity, as it is necessary to solve a linear regression with $n^2$ variable and $n^{\delta(e)}$ observations.

In the recent work on network motif clustering [9], the hyperedges are deduced from a graph where they represent so called motifs. Benson et. al [9] proved that if the motifs have three vertices, resulting in a three-uniform hypergraph, their proposed algorithm satisfies the Cheeger inequality for motifs[2]. In the described formulation, when cutting an H-hyperedge with weight $w_e^{\mathrm{H}}$, one is required to pay $w_e^{\mathrm{H}}$. Hence, recasting this model within our setting, we arrive at inhomogenous weights $w_e(S) = w_e^{\mathrm{H}}$, for all $S \in 2^e$, for which (5) yields $w_{v\tilde{v}}^{(e)} = w_e^{\mathrm{H}}/(\delta(e) - 1)$ and $\beta^{(e)} = \lfloor \frac{\delta^2(e)}{4} \rfloor/(\delta(e) - 1)$,

identical to the solution of [9]. Furthermore, given the result of our Theorem 3.1, one can prove that the algorithm of [9] offers a quadratic-factor approximation for motifs involving more than three vertices, a fact that was not established in the original work [9].

All the aforementioned algorithms essentially learn the spectrum of Laplacian matrices obtained through hypergraph projection. The ultimate goal of projections is to avoid solving the NP-hard problem of learning the spectrum of certain hypergraph Laplacians [24]. Methods that do not rely on hypergraph projection, including optimization with the total variance of hypergraphs [12, 13], tensor spectral methods [25] and nonlinear Laplacian spectral methods [26], have also been reported in the literature. These techniques were exclusively applied in homogeneous settings, and they typically have higher complexity and smaller spectral gaps than the projection-based methods. A future line of work is to investigate whether these methods can be extended to the inhomogeneous case. Yet another relevant line of work pertains to the statistical analysis of hypergraph partitioning methods for generalized stochastic block models [27, 28].

## 5 Applications

**Network motif clustering.** Real-world networks exhibit rich higher-order connectivity patterns frequently referred to as network motifs [29]. Motifs are special subgraphs of the graph and may be viewed as hyperedges of a hypergraph over the same set of vertices. Recent work has shown that hypergraph clustering based on motifs may be used to learn hidden high-order organization patterns in networks [9, 8, 21]. However, this approach treats all vertices and edges within the motifs in the same manner, and hence ignores the fact that each structural unit within the motif may have a different relevance or different role. As a result, the vertices of the motifs are partitioned with a uniform cost. However, this assumption is hardly realistic as in many real networks, only some vertices of higher-order structures may need to be clustered together. Hence, inhomogenous hyperedges are expected to elucidate more subtle high-order organizations of network. We illustrate the utility of InH-partition on the Florida Bay foodweb [30] and compare our findings to those of [9].

The Florida Bay foodweb comprises 128 vertices corresponding to different species or organisms that live in the Bay, and 2106 directed edges indicating carbon exchange between two species. The Foodweb essentially represents a layered flow network, as carbon flows from so called producers organisms to high-level predators. Each layer of the network consists of "similar" species that play the same role in the food chain. Clustering of the species may be performed by leveraging the layered structure of the interactions. As a network motif, we use a subgraph of four species, and correspondingly, four vertices denoted by $v_i$, for $i = 1, 2, 3, 4$. The motif captures, among others, relations between two producers and two consumers: The producers $v_1$ and $v_2$ both transmit carbons to $v_3$ and $v_4$, and all types of carbon flow between $v_1$ and $v_2$, $v_3$ and $v_4$ are allowed (see Figure 2 Left). Such a motif is the smallest structural unit that captures the fact that carbon exchange occurs in uni-direction between layers, while is allowed freely within layers. The inhomogeneous hyperedge costs are assigned according to the following heuristics: First, as $v_1$ and $v_2$ share two common carbon recipients (predators) while $v_3$ and $v_4$ share two common carbon sources (preys), we set $w_e(\{v_i\}) = 1$ for $i = 1, 2, 3, 4$, and $w_e(\{v_1, v_2\}) = 0$, $w_e(\{v_1, v_3\}) = 2$, and $w_e(\{v_1, v_4\}) = 2$. Based on the solution of the optimization problem (5), one can construct a weighted subgraph whose costs of cuts match the inhomogeneous costs, with $\beta^{(e)} = 1$. The graph is depicted in Figure 2 (left).

Our approach is to perform hierarchical clustering via iterative application of the InH-partition method. In each iteration, we construct a hypergraph by replacing the chosen motif subnetwork by an hyperedge. The result is shown in Figure 2. At the first level, we partitioned the species into three clusters corresponding to producers, primary consumers and secondary consumers. The producer cluster is homogeneous in so far that it contains only producers, a total of nine of them. At the second level, we partitioned the obtained primary-consumer cluster into two clusters, one of which almost exclusively comprises invertebrates (28 out of 35), while the other almost exclusively comprises forage fishes. The secondary-consumer cluster is partitioned into two clusters, one of which comprises top-level predators, while the other cluster mostly consists of predatory fishes and birds. Overall, we recovered five clusters that fit five layers ranging from producers to top-level consumers. It is easy to check that the producer, invertebrate and top-level predator clusters exhibit high functional similarity of species ($> 80\%$). An exact functional classification of forage and predatory fishes is not known, but our layered network appears to capture an overwhelmingly large number of prey-predator relations among these species. Among the 1714 edges, obtained after removing isolated vertices and detritus species vertices, only five edges point in the opposite direction from a higher to a lower-level

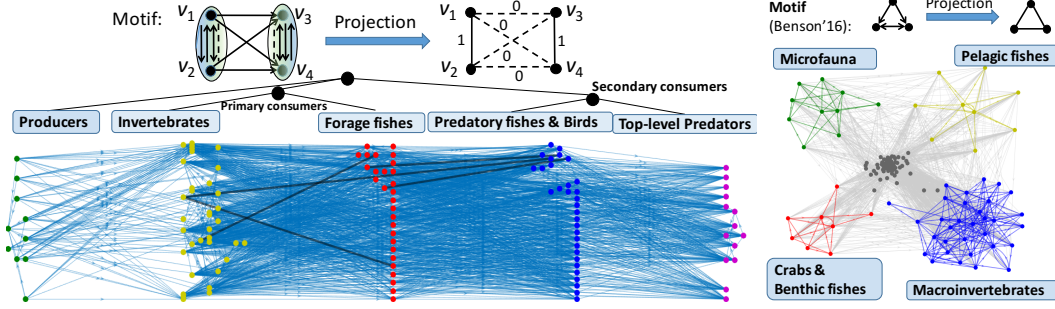

Figure 2: Motif clustering in the Florida Bay food web. Left: InHomogenous case. Left-top: Hyperedge (network motif) & the weighted induced subgraph; Left-bottom: Hierarchical clustering structure and five clusters via InH-partition. The vertices belonging to different clusters are distinguished by the colors of vertices. Edges with a uni-direction (right to left) are colored black while other edges are kept blue. Right: Homogenous partitioning [9] with four clusters. Grey vertices are not connected by motifs and thus unclassified.

cluster, two of which go from predatory fishes to forage fishes. Detailed information about the species and clusters is provided in the Supplementary Material.

In comparison, the related work of Benson et al. [9] which used homogenous hypergraph clustering and triangular motifs reported a very different clustering structure. The corresponding clusters covered less than half of the species (62 out of 128) as many vertices were not connected by the triangle motif; in contrast, 127 out of 128 vertices were covered by our choice of motif. We attribute the difference between our results and the results of [9] to the choices of the network motif. A triangle motif, used in [9] leaves a large number of vertices unclustered and fails to enforce a hierarchical network structure. On the other hand, our fan motif with homogeneous weights produces a giant cluster as it ties all the vertices together, and the hierarchical decomposition is only revealed when the fan motif is used with inhomogeneous weights. In order to identify hierarchical network structures, instead of hypergraph clustering, one may use topological sorting to rank species based on their carbon flows [31]. Unfortunately, topological sorting cannot use biological side information and hence fails to automatically determine the boundaries of the clusters.

**Learning the Riffled Independence Structure of Ranking Data.** Learning probabilistic models for ranking data has attracted significant interest in social and political sciences as well as in machine learning [32, 33]. Recently, a probabilistic model, termed the riffled-independence model, was shown to accurately describe many benchmark ranked datasets [34]. In the riffled independence model, one first generates two rankings over two disjoint sets of element independently, and then riffle shuffles the rankings to arrive at an interleaved order. The structure learning problem in this setting reduces to distinguishing the two categories of elements based on limited ranking data. More precisely, let $Q$ be the set of candidates to be ranked, with $|Q| = n$. A full ranking is a bijection $\sigma : Q \to [n]$, and for an $a \in Q$, $\sigma(a)$ denotes the position of candidate $a$ in the ranking $\sigma$. We use $\sigma(a) < (>)\sigma(b)$ to indicate that $a$ is ranked higher (lower) than $b$ in $\sigma$. If $S \subseteq Q$, we use $\sigma_S : S \to [|S|]$ to denote the ranking $\sigma$ projected onto the set $S$. We also use $S(\sigma) \triangleq \{\sigma(a)|a \in S\}$ to denote the subset of positions of elements in $S$. Let $\mathbb{P}(E)$ denote the probability of the event $E$. Riffled independence asserts that there exists a riffled-independent set $S \subset Q$, such that for a fixed ranking $\sigma'$ over $[n]$,

$$\mathbb{P}(\sigma = \sigma') = \mathbb{P}(\sigma_S = \sigma'_S)\mathbb{P}(\sigma_{Q/S} = \sigma'_{Q/S})\mathbb{P}(S(\sigma) = S(\sigma')).$$

Suppose that we are given a set of rankings $\Sigma = \{\sigma^{(1)}, \sigma^{(2)}, ..., \sigma^{(m)}\}$ drawn independently according to some probability distribution $\mathbb{P}$. If $\mathbb{P}$ has a riffled-independent set $S^*$, the structure learning problem is to find $S^*$. In [34], the described problem was cast as an optimization problem over all possible subsets of $Q$, with the objective of minimizing the Kullback-Leibler divergence between the ranking distribution with riffled independence and the empirical distribution of $\Sigma$ [34]. A simplified version of the optimization problem reads as

$$\arg\min_{S \subset Q} \mathcal{F}(S) \triangleq \sum_{(i,j,k) \in \Omega^{cross}_{S,\bar{S}}} I_{i;j,k} + \sum_{(i,j,k) \in \Omega^{cross}_{\bar{S},S}} I_{i;j,k}, \tag{12}$$

where $\Omega^{cross}_{A,B} \triangleq \{(i,j,k)|i \in A, j,k \in B\}$, and where $I_{i;j,k}$ denotes the estimated mutual information between the position of the candidate $i$ and two "comparison candidates" $j, k$. If $1_{\sigma(j)<\sigma(k)}$

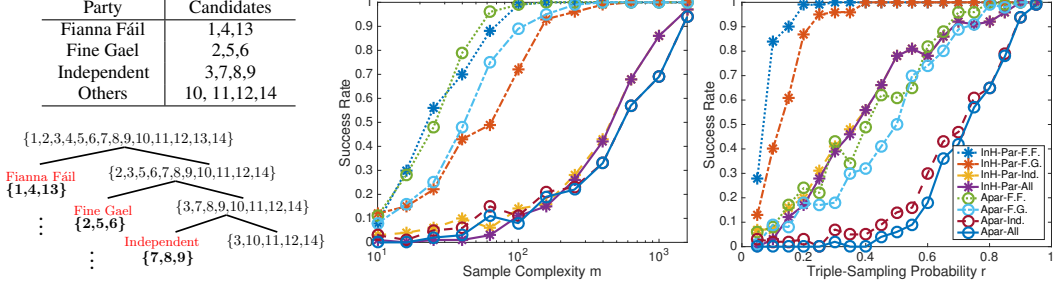

Figure 3: Election dataset. Left-top: parties and candidates; Left-bottom: hierarchical partitioning structure of Irish election detected by InH-Par; Middle: Success rate vs Sample Complexity; Right: Success rate vs Triple-sampling Rate.

denotes the indicator function of the underlying event, we may write

$$I_{i;j,k} \triangleq \hat{I}(\sigma(i); 1_{\sigma(j)<\sigma(k)}) = \sum_{\sigma(i)} \sum_{1_{\sigma(j)<\sigma(k)}} \hat{\mathbb{P}}(\sigma(i), 1_{\sigma(j)<\sigma(k)}) \log \frac{\hat{\mathbb{P}}(\sigma(i), 1_{\sigma(j)<\sigma(k)})}{\hat{\mathbb{P}}(\sigma(i))\mathbb{P}(1_{\sigma(j)<\sigma(k)})}, \quad (13)$$

where $\hat{\mathbb{P}}$ denotes an estimate of the underlying probability. If $i$ and $j, k$ are in different riffled-independent sets, the estimated mutual information $\hat{I}(\sigma(i); 1_{\sigma(j)<\sigma(k)})$ converges to zero as the number of samples increases. When the number of samples is small, one may use mutual information estimators described in [35, 36, 37].

One may recast the above problem as an InH-partition problem over a hypergraph where each candidate represents a vertex in the hypergraph, and $I_{i;j,k}$ represents the inhomogeneous cost $w_e(\{i\})$ for the hyperedge $e = \{i, j, k\}$. Note that as mutual information $\hat{I}(\sigma(i); 1_{\sigma(j)<\sigma(k)})$ is in general asymmetric, one would not have been able to use H-partitions. The optimization problem reduces to $\min_S \text{vol}_{\mathcal{H}}(\partial S)$. The two optimization tasks are different, and we illustrate next that the InH-partition outperforms the original optimization approach AnchorsPartition (Apar) [34] both on synthetic data and real data. Due to space limitations, synthetic data and a subset of the real dataset results are listed in the Supplementary Material.

Here, we analyzed the Irish House of Parliament election dataset (2002) [38]. The dataset consists of 2490 ballots fully ranking 14 candidates. The candidates were from a number of parties, where Fianna Fáil (F.F.) and Fine Gael (F.G.) are the two largest (and rival) Irish political parties. Using InH-partition (InH-Par), one can split the candidates iteratively into two sets (See Figure 3) which yields to meaningful clusters that correspond to large parties: $\{1, 4, 13\}$ (F.F.), $\{2, 5, 6\}$ (F.G.), $\{7, 8, 9\}$ (Ind.). We compared InH-partition with Apar based on their performance in detecting these three clusters using a small training set: We independently sampled $m$ rankings 100 times and executed both algorithms to partition the set of candidates iteratively. During the partitioning procedure, "party success" was declared if one exactly detected one of the three party clusters ("F.F.", "F.G." & "Ind."). "All" was used to designate that all three party clusters were detected completely correctly. InH-partition outperforms Apar in recovering the cluster Ind. and achieved comparable performance for cluster F.F., although it performs a little worse than Apar for cluster F.G.; InH-partition also offers superior overall performance compared to Apar. We also compared InH-partition with APar in the large sample regime ($m = 2490$), using only a subset of triple comparisons (hyperedges) sampled independently with probability $r$ (This strategy significantly reduces the complexity of both algorithms). The average is computed over 100 independent runs. The results are shown in Figure 3, highlighting the robustness of InH-partition with respect to missing triples. Additional test on ranking data are described in the Supplementary Material, along with new results on subspace clustering, motion segmentation and others.

## 6 Acknowledgement

The authors gratefully acknowledge many useful suggestions by the reviewers. They are also indebted to the reviewers for providing many additional and relevant references. This work was supported in part by the NSF grant CCF 1527636.

## Footnotes

[1]The code for experiments can be found at https://github.com/lipan00123/InHclustering.

[2]The Cheeger inequality [17] arises in the context of minimizing the conductance of a graph, which is related to the normalized cut.

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
