[Supplementary Material]

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

## A  Conductance of a Cut for an Inhomogeneous Hypergraph

The conductance of a cut $(S, \bar{S})$ for an inhomogeneous hypergraph is defined according to

$$\psi_{\mathcal{H}}(S) = \frac{\text{vol}_{\mathcal{H}}(\partial S)}{\min\{\text{vol}_{\mathcal{H}}(S), \text{vol}_{\mathcal{H}}(\bar{S})\}}. \tag{14}$$

This definition is consistent with the definition of conductance of a graph cut [39] and homogeneoous hypergraph cut [9]. The conductance is an upper bound for the normalized cut

$$\psi_{\mathcal{H}}(S) \geq \frac{1}{2}\text{NCut}_{\mathcal{H}}(S). \tag{15}$$

## B  A Brief Overview of Spectral Graph Partitioning

The combinatorial optimization problem of minimizing the NCut (3) for graphs is known to be NP-complete [18]. However, an efficient algorithm based on spectral techniques (Algorithm 1 below) can produce a solution within a quadratic factor of the optimum (Theorem. B.1).

---
**Algorithm 1: Spectral Graph Partitioning**
**Input:** The adjacency matrix $A$ of the graph $\mathcal{G}$.
 Step 1: Construct the diagonal degree matrix $D$, with $D_{ii} = \sum_{j=1}^{n} A_{ij}$ for all $i \in [n]$.
 Step 2: Construct the normalized Laplacian matrix $\mathcal{L} = I - D^{-1/2}AD^{-1/2}$.
 Step 3: Compute the eigenvector $\mathbf{u} = (u_1, u_2, ..., u_n)^T$ corresponding to the second smallest eigenvalue of $\mathcal{L}$.
 Step 4: Let $\ell_i$ be the index of the $i$-th smallest entry of $D^{-1/2}\mathbf{u}$.
 Step 5: Compute $S = \arg\min_{S_i, 1 \leq i \leq n-1} \text{NCut}_{\mathcal{G}}(S_i)$ over all sets $S_i = \{\ell_1, \ell_2, \ldots, \ell_i\}$.
**Output:** Output $S$ if $|S| < |\bar{S}|$, and $\bar{S}$ otherwise.

---

**Theorem B.1.** [**Derived based on Theorem 1 [17]**] Let $\alpha$ denote the value of the NCut output by Algorithm 1, and let $\alpha_{\mathcal{G}}$ denotes the optimal NCut of the graph $\mathcal{G}$. Also, let $\lambda_{\mathcal{G}}$ stand for the second smallest eigenvalue of normalized Laplacian matrix $\mathcal{L}$. Then,

$$\alpha_{\mathcal{G}} \geq \lambda_{\mathcal{G}} \geq \frac{\alpha^2}{8} \geq \frac{\alpha_{\mathcal{G}}^2}{8}. \tag{16}$$

*Proof.* Let $S^*$ denote the output of Algorithm 1. We have $\alpha_{\mathcal{G}} \leq \text{NCut}_{\mathcal{G}}(S^*) = \alpha$ based on the definition of $\alpha_{\mathcal{G}}$. Suppose that $\alpha_{\mathcal{G}}$ is achieved by the set $S_o$, i.e., that

$$\alpha_{\mathcal{G}} = \text{vol}_{\mathcal{G}}(\partial S_o) \left( \frac{1}{\text{vol}_{\mathcal{G}}(S_o)} + \frac{1}{\text{vol}_{\mathcal{G}}(\bar{S}_o)} \right).$$

Let $V$ be the vertices of $\mathcal{G}$. We define the following function $g$ over $V$

$$g(v) = 1_{v \in S_o} - \frac{\text{vol}_{\mathcal{G}}(S_o)}{\text{vol}_{\mathcal{G}}(V)}, \quad \forall \, v \in V.$$

Recall that $d_v$ is the degree of vertex $v$. It is easy to check that $\sum_{v \in V} g(v)d_v = 0$. Hence, due to the definition of $\lambda_{\mathcal{G}}$, we have

$$\lambda_{\mathcal{G}} \leq \frac{g^T(D - A)g}{g^T Dg} = \alpha_{\mathcal{G}},$$

where $A$ denotes the adjacency matrix of the graph and $D$ is a diagonal matrix whose diagonal entries equal to the degrees of the corresponding vertices. According to Theorem 1 [17] and the results of (15), we have

$$\lambda_{\mathcal{G}} \geq \frac{\psi_{\mathcal{G}}^2(S^*)}{2} \geq \frac{\alpha^2}{8},$$

which concludes the proof. $\square$

For $k$-way partitions, Step 3 of the previously described algorithm entails computing the eigenvectors corresponds to the $k$ smallest eigenvalues of $\mathcal{L}$. The $i$-th components of these $k$ eigenvectors may be viewed as the coordinates of a representation for vertex $v_i$. Partitioning is performed by using the eigenvector representations of the vertices in some distance-based clustering algorithms, such as $k$-means.

## C Proof of Theorem 3.1

Assume that the optimization problem (5) is feasible for all InH-hyperedges, and recall that $\beta^* = \max_{e \in E} \beta^{(e)}$. Then $\mathcal{G} = (V, E_o, w)$ can constantly approximate the original hypergraph $\mathcal{H}$ in the sense that

$$\text{Vol}_{\mathcal{H}}(\partial S) \le \text{Vol}_{\mathcal{G}}(\partial S) \le \beta^* \text{Vol}_{\mathcal{H}}(\partial S), \quad \text{Vol}_{\mathcal{H}}(S) \le \text{Vol}_{\mathcal{G}}(S) \le \beta^* \text{Vol}_{\mathcal{H}}(S). \qquad (17)$$

Furthermore, recall that $w_{v\tilde{v}}$ is the weight of the edge $v\tilde{v}$ of the graph $\mathcal{G}$. If the weights $w_{v\tilde{v}}$ are non-negative for all $\{v, \tilde{v}\} \in E_o$, we can use Theorem B.1 when performing Algorithm 1 over the graph $\mathcal{G}$. Combining Theorem B.1 and equation (17), we have

$$(\beta^*)^3 \alpha_{\mathcal{H}} \ge \frac{(\alpha^*)^2}{8} \ge \frac{\alpha_{\mathcal{H}}^2}{8}, \qquad (18)$$

which concludes the proof.

## D Proof of Theorem 3.3

First, recall that $w_{v\tilde{v}}^{*(e)}$ denotes the projection weight of equation (10), for the case that $w_e(\cdot)$ is submodular:

$$w_{v\tilde{v}}^{*(e)} = \sum_{S \in 2^e/\{\emptyset, e\}} \left[ \frac{w_e(S)}{2|S|(\delta(e) - |S|)} 1_{|\{v,\tilde{v}\} \cap S|=1} \right. \qquad (19)$$

$$\left. - \frac{w_e(S)}{2(|S|+1)(\delta(e) - |S| - 1)} 1_{|\{v,\tilde{v}\} \cap S|=0} - \frac{w_e(S)}{2(|S|-1)(\delta(e) - |S| + 1)} 1_{|\{v,\tilde{v}\} \cap S|=2} \right]$$

We start by proving that for a fixed pair of vertices $v$ and $\tilde{v}$, the weights $w_{v\tilde{v}}^{*(e)}$ are nonnegative provided that the $w_e(\cdot)$ are submodular. Note that the sum on the right-hand side of (19) is over all proper subsets $S$. The coefficients of $w_e(S)$ are positive if and only if $S$ contains exactly one of the endpoints $v$ and $\tilde{v}$. The idea behind the proof is to construct bijections between the subsets with positive coefficients and those with negative coefficients and cancel negative and positive terms.

We partition the power set $2^e$ into four parts, namely

$$\mathbb{S}_1 \triangleq \{S \in 2^e : v \in S, \tilde{v} \notin S\},$$
$$\mathbb{S}_2 \triangleq \{S \in 2^e : v \notin S, \tilde{v} \in S\},$$
$$\mathbb{S}_3 \triangleq \{S \in 2^e : v \notin S, \tilde{v} \notin S\},$$
$$\mathbb{S}_4 \triangleq \{S \in 2^e : v \in S, \tilde{v} \in S\}.$$

Choose any $S_1 \in \mathbb{S}_1$ and construct the unique sets $S_2 = S_1/\{v\} \cup \{\tilde{v}\} \in \mathbb{S}_2$, $S_3 = S_1/\{v\} \in \mathbb{S}_3$, $S_4 = S_1 \cup \{\tilde{v}\} \in \mathbb{S}_4$. Consequently, each set may be reconstructed from another set in the group, and we denote this set of bijective relations by $S_1 \leftrightarrow S_2 \leftrightarrow S_3 \leftrightarrow S_4$. Let $s = |S_1|$. Due to the way the sets $S_1$ and $S_2$ are chosen, the corresponding coefficients of $w_e(S_1)$ and $w_e(S_2)$ in (10) are both equal to

$$\frac{1}{2s(\delta(e) - s)}.$$

We also observe that the corresponding coefficients of $w_e(S_3)$ and $w_e(S_4)$ are

$$-\frac{1}{2s(\delta(e) - s)}.$$

Note that the submodularity property

$$w_e(S_1) + w_e(S_2) \ge w_e(S_3) + w_e(S_4),$$

allows us to cancel out the negative terms in the sum (19). This proves the claimed result.

Next, we prove that the optimization problem (5) has a feasible solution.

Recall that $G_e = (V^{(e)}, E^{(e)}, w^{(e)})$ is the subgraph obtained by projecting $e$. Set $w^{(e)} = w^{*(e)}$. For simplicity of notation, we denote the volume of the boundary of $S$ over $G_e$ as

$$\text{Vol}_{G_e}(\partial S) = \sum_{v \in S, \tilde{v} \in e/S} w_{v\tilde{v}}^{(e)}, \qquad \text{for } S \in 2^e.$$

The existence of a feasible solution of the optimization problem may be verified by checking that for any $S \in 2^e/\{\emptyset, e\}$, and for a given $\beta^{(e)}$, we have the following bounds on the volume of the boundary of $S$:

$$w_e(S) \leq \text{Vol}_{G_e}(\partial S) \leq \beta^{(e)} w_e(S).$$

Due to symmetry, we only need to perform the verification for sets $S$ of different cardinalities $|S| \leq \delta(e)/2$. This verification is performed on a case-by-case bases, as we could not establish a general proof for arbitrary degree $\delta(e) \geq 2$. In what follows, we show that the claim holds true for all $\delta(e) \leq 7$; based on several special cases considered, we conjecture that the result is also true for all values of $\delta(e)$ greater than seven.

For notational simplicity, we henceforth assume that the vertices in $e$ are labeled by elements in $\{1, 2, 3, ..., \delta(e)\}$.

First, note that by combining symmetry and submodularity, we can easily show that

$$w_e(S_1) + w_e(S_2) = w_e(S_1) + w_e(\bar{S}_2) \geq w_e(S_1 \cup \bar{S}_2) + w_e(S_1 \cap \bar{S}_2) = w_e(S_2/S_1) + w_e(S_1/S_2).$$

We iteratively use this equality in our subsequent proofs, following a specific notational format for all relevant inequalities:

$$v_{i_1}, ..., v_{i_r} \in S_1, \ v_{j_1}, ..., v_{j_s} \in S_2, \ \textit{Weight inequality} \implies \textit{Volume inequality}.$$

The above line asserts that for all ordered subsets $(v_{i_1}, ..., v_{i_r})$ and $(v_{j_1}, ..., v_{j_s})$ chosen from $S_1$ and $S_2$ without replacement, respectively, we have that the *Weight inequalities* follow based on the properties of $w_e(\cdot)$. These *Weight inequalities* are consequently inserted into the formula for the volume $\text{Vol}_{G_e}(S)$ to arrive at the *Volume inequality* for $\text{Vol}_{G_e}(S)$.

For $\delta(e) = 2$, the projection (19) is just a "self-projection": It is easy to check that for any singleton $S$, $\text{Vol}_{G_e}(\partial S) = w_e(S)$ and hence $\beta^{(e)} = 1$. We next establish the same claim for larger hyperedge sizes $\delta(e)$.

**D.1** $\delta(e) = 3$, $\beta^{(e)} = 1$

By using the symmetry property of $w_e(\cdot)$ we have

$$w_{12}^{*(e)} = \frac{1}{2}(w_e(\{1\}) + w_e(\{2\})) - w_e(\{3\}).$$

Therefore, $\text{Vol}_{G_e}(\partial\{1\}) = w_{12}^{*(e)} + w_{13}^{*(e)} = w_e(\{1\})$ and hence $\beta^{(e)} = 1$.

**D.2** $\delta(e) = 4$, $\beta^{(e)} = 3/2$

By using the symmetry property of $w_e(\cdot)$ we have

$$w_{12}^{*(e)}$$
$$= \frac{1}{3}(w_e(\{1\}) + w_e(\{2\})) - \frac{1}{4}(w_e(\{3\}) + w_e(\{4\}))$$
$$+ \frac{1}{4}w_e(\{1,3\}) + w_e(\{1,4\})) - \frac{1}{3}w_e(\{1,2\}).$$

The basic idea behind the proof of the equalities to follow is to carefully select subsets for which the submodular inequality involving $w_e(\cdot)$ may be used to eliminate the terms corresponding to the volumes $\text{Vol}_{G_e}(\partial S)$.

**Case $S = \{1\}$:**

$$\text{Vol}_{G_e}(\partial\{1\})$$
$$= w_e(\{1\}) - \frac{1}{6}(w_e(\{2\}) + w_e(\{3\}) + w_e(\{4\}))$$
$$+ \frac{1}{6}(w_e(\{1,2\}) + w_e(\{1,3\}) + w_e(\{1,4\}))$$

$$v_1 = 1, \ v_2, v_3 \in \{2,3,4\}, \ w_e(\{v_1, v_2\}) + w_e(\{v_1, v_3\}) \geq w_e(\{v_2\}) + w_e(\{v_3\})$$
$$\implies \quad \text{Vol}_{G_e}(\partial\{1\}) \geq w_e(\{1\}).$$

$$v_1 = 1, \ v_2 \in \{2,3,4\}, \ w_e(\{v_1, v_2\}) \leq w_e(\{v_1\}) + w_e(\{v_2\})$$
$$\implies \quad \text{Vol}_{G_e}(\partial\{1\}) \leq \frac{3}{2} w_e(\{1\}).$$

**Case $S = \{1,2\}$:**

$$\text{Vol}_{G_e}(\partial\{1,2\})$$
$$= \frac{1}{6}(w_e(\{1\}) + w_e(\{2\}) + w_e(\{3\}) + w_e(\{4\}))$$
$$+ w_e(\{1,2\}) - \frac{1}{6}(w_e(\{1,3\}) + w_e(\{1,4\}))$$

$$v_1 \in \{1,2\}, \ v_2 \in \{3,4\}, \ w_e(\{v_1\}) + w_e(\{v_2\}) \geq w_e(\{v_1, v_2\})$$
$$\implies \quad \text{Vol}_{G_e}(\partial\{1,2\}) \geq w_e(\{1,2\}).$$

$$v_1, v_2 \in \{1,2\}, v_3 \in \{3,4\}, \ w_e(\{v_1\}) + w_e(\{v_3\}) \leq w_e(\{v_1, v_2\}) + w_e(\{v_2, v_3\})$$
$$\implies \quad \text{Vol}_{G_e}(\partial\{1,2\}) \leq \frac{4}{3} w_e(\{1,2\}).$$

**D.3** $\quad \delta(e) = 5, \ \beta^{(e)} = 2$

By using the symmetry property of $w_e(\cdot)$ we have

$$w_{12}^{*(e)}$$
$$= \frac{1}{4}(w_e(\{1\}) + w_e(\{2\})) - \frac{1}{6}(w_e(\{3\}) + w_e(\{4\}) + w_e(\{5\})) - \frac{1}{4}w_e(\{1,2\})$$
$$+ \frac{1}{6}(w_e(\{1,3\}) + w_e(\{1,4\}) + w_e(\{1,5\}) + w_e(\{2,3\}) + w_e(\{2,4\}) + w_e(\{2,5\}))$$
$$- \frac{1}{6}(w_e(\{3,4\}) + w_e(\{3,5\}) + w_e(\{4,5\})).$$

**Case $S = \{1\}$:**

$$\text{Vol}_{G_e}(\partial\{1\})$$
$$= w_e(\{1\}) - \frac{1}{4}(w_e(\{2\}) + w_e(\{3\}) + w_e(\{4\}) + w_e(\{5\}))$$
$$+ \frac{1}{4}(w_e(\{1,2\}) + w_e(\{1,3\}) + w_e(\{1,4\}) + w_e(\{1,5\}))$$

$$v_1 = 1, \ v_2, v_3 \in \{2,3,4,5\}, \ w_e(\{v_1, v_2\}) + w_e(\{v_1, v_3\}) \geq w_e(\{v_2\}) + w_e(\{v_3\})$$
$$\implies \quad \text{Vol}_{G_e}(\partial\{1\}) \geq w_e(\{1\}).$$

$$v_1 = 1, \ v_2 \in \{2,3,4,5\}, \ w_e(\{v_1, v_2\}) \leq w_e(\{v_1\}) + w_e(\{v_2\})$$
$$\implies \quad \text{Vol}_{G_e}(\partial\{1\}) \leq 2w_e(\{1\}).$$

**Case $S = \{1, 2\}$:**

$$\text{Vol}_{G_e}(\partial\{1, 2\})$$

$$=\frac{1}{4}(w_e(\{1\}) + w_e(\{2\})) - \frac{1}{6}(w_e(\{3\}) + w_e(\{4\}) + w_e(\{5\})) + w_e(\{1, 2\})$$

$$-\frac{1}{12}(w_e(\{1, 3\}) + w_e(\{1, 4\}) + w_e(\{1, 5\}) + w_e(\{2, 3\}) + w_e(\{2, 4\}) + w_e(\{2, 5\}))$$

$$+\frac{1}{3}(w_e(\{3, 4\}) + w_e(\{3, 5\}) + w_e(\{4, 5\}))$$

$v_1, v_2, v_3 \in \{3, 4, 5\}, \ w_e(\{v_2\}) + w_e(\{v_3\}) \leq w_e(\{v_1, v_2\}) + w_e(\{v_1, v_3\})$
$v_1 \in \{1, 2\}, \ v_2 \in \{3, 4, 5\}, \ w_e(\{v_1, v_2\}) \leq w_e(\{v_1\}) + w_e(\{v_2\})$
$$\implies \quad \text{Vol}_{G_e}(\partial\{1, 2\}) \geq w_e(\{1, 2\}).$$

$v_1 \in \{1, 2\}, \ v_2, v_3 \in \{3, 4, 5\}, \ w_e(\{v_1, v_2\}) + w_e(\{v_1, v_3\}) \geq w_e(\{v_2\}) + w_e(\{v_3\})$
$v_1, v_2 \in \{1, 2\}, \ v_3, v_4, v_5 \in \{3, 4, 5\}, \ w_e(\{v_3, v_4\}) \leq w_e(\{v_1, v_2\}) + w_e(\{v_5\})$
$$\implies \quad \text{Vol}_{G_e}(\partial\{1, 2\}) \leq 2w_e(\{1, 2\}).$$

**D.4**  $\delta(e) = 6$, $\beta^{(e)} = 4$

By using the symmetry property of $w_e(\cdot)$ we have

$$w_{12}^{*(e)}$$

$$=\frac{1}{5}(w_e(\{1\}) + w_e(\{2\})) - \frac{1}{8}(w_e(\{3\}) + w_e(\{4\}) + w_e(\{5\}) + w_e(\{6\})) - \frac{1}{5}w_e(\{1, 2\})$$

$$+\frac{1}{8}(w_e(\{1, 3\}) + w_e(\{1, 4\}) + w_e(\{1, 5\}) + w_e(\{1, 6\}) + w_e(\{2, 3\}) + w_e(\{2, 4\})$$

$$+ w_e(\{2, 5\}) + w_e(\{2, 6\}))$$

$$-\frac{1}{9}(w_e(\{3, 4\}) + w_e(\{3, 5\}) + w_e(\{3, 6\}) + w_e(\{4, 5\}) + w_e(\{4, 6\}) + w_e(\{5, 6\}))$$

$$-\frac{1}{9}(w_e(\{1, 2, 3\}) + w_e(\{1, 2, 4\}) + w_e(\{1, 2, 5\}) + w_e(\{1, 2, 6\}))$$

$$+\frac{1}{8}(w_e(\{1, 3, 4\}) + w_e(\{1, 3, 5\}) + w_e(\{1, 3, 6\}) + w_e(\{1, 4, 5\})$$

$$+ w_e(\{1, 4, 6\}) + w_e(\{1, 5, 6\}))$$

**Case $S = \{1\}$:**

$$\text{Vol}_{G_e}(\partial\{1\})$$

$$=w_e(\{1\}) - \frac{3}{10}(w_e(\{2\}) + w_e(\{3\}) + w_e(\{4\}) + w_e(\{5\}) + w_e(\{6\}))$$

$$+\frac{3}{10}(w_e(\{1, 2\}) + w_e(\{1, 3\}) + w_e(\{1, 4\}) + w_e(\{1, 5\}) + w_e(\{1, 6\}))$$

$$-\frac{1}{12}(w_e(\{2, 3\}) + w_e(\{2, 4\}) + w_e(\{2, 5\}) + w_e(\{2, 6\}) + w_e(\{3, 4\})$$

$$+ w_e(\{3, 5\}) + w_e(\{3, 6\}) + w_e(\{4, 5\}) + w_e(\{4, 6\}) + w_e(\{5, 6\}))$$

$$+\frac{1}{12}(w_e(\{1, 2, 3\}) + w_e(\{1, 2, 4\}) + w_e(\{1, 2, 5\}) + w_e(\{1, 2, 6\})$$

$$+ w_e(\{1, 3, 4\}) + w_e(\{1, 3, 5\}) + w_e(\{1, 3, 6\})$$

$$+ w_e(\{1, 4, 5\}) + w_e(\{1, 4, 6\}) + w_e(\{1, 5, 6\}))$$

$$v_1 = 1, \ v_2, v_3 \in \{2,3,4,5,6\}, \ w_e(\{v_1,v_2\}) + w_e(\{v_1,v_3\}) \geq w_e(\{v_2\}) + w_e(\{v_3\})$$
$$v_1 = 1, \ v_2, v_3, v_4, v_5 \in \{2,3,4,5,6\},$$
$$w_e(\{v_1,v_2,v_3\}) + w_e(\{v_1,v_4,v_5\}) \geq w_e(\{v_2,v_3\}) + w_e(\{v_4,v_5\})$$
$$\implies \mathrm{Vol}_{G_e}(\partial\{1\}) \geq w_e(\{1\}).$$

$$v_1 = 1, \ v_2 \in \{2,3,4,5,6\}, \ w_e(\{v_1,v_2\}) \leq w_e(\{v_1\}) + w_e(\{v_2\})$$
$$v_1 = 1, \ v_2, v_3 \in \{2,3,4,5,6\}, \ w_e(\{v_1,v_2,v_3\}) \leq w_e(\{v_1\}) + w_e(\{v_2,v_3\})$$
$$\implies \quad \mathrm{Vol}_{G_e}(\partial\{1\}) \leq \frac{10}{3} w_e(\{1\}).$$

**Case $S = \{1,2\}$:**

$\mathrm{Vol}_{G_e}(\partial\{1,2\})$

$$= \frac{3}{10}(w_e(\{1\}) + w_e(\{2\})) - \frac{7}{20}(w_e(\{3\}) + w_e(\{4\}) + w_e(\{5\}) + w_e(\{6\}))$$

$$+ w_e(\{1,2\}) - \frac{1}{30}(w_e(\{1,3\}) + w_e(\{1,4\}) + w_e(\{1,5\}) + w_e(\{1,6\}) + w_e(\{2,3\})$$
$$+ w_e(\{2,4\}) + w_e(\{2,5\}) + w_e(\{2,6\}))$$

$$+ \frac{1}{18}(w_e(\{3,4\}) + w_e(\{3,5\}) + w_e(\{3,6\}) + w_e(\{4,5\}) + w_e(\{4,6\}) + w_e(\{5,6\}))$$

$$+ \frac{5}{12}(w_e(\{1,2,3\}) + w_e(\{1,2,4\}) + w_e(\{1,2,5\}) + w_e(\{1,2,6\}))$$

$$- \frac{1}{18}(w_e(\{1,3,4\}) + w_e(\{1,3,5\}) + w_e(\{1,3,6\}) + w_e(\{1,4,5\}) + w_e(\{1,4,6\}) + w_e(\{1,5,6\}))$$

$$v_1 \in \{1,2\}, \ v_2 \in \{3,4,5,6\}, \ w_e(\{v_1,v_2\}) \leq w_e(\{v_1\}) + w_e(\{v_2\})$$
$$v_1 = 1, \ v_2 = 2, \ v_3, v_4 \in \{3,4,5,6\}, \ w_e(\{v_3\}) + w_e(\{v_4\}) \leq w_e(\{v_1,v_2,v_3\}) + w_e(\{v_1,v_2,v_4\})$$
$$v_1 = 1, \ v_2, v_3 \in \{3,4,5,6\}, \ w_e(\{v_1,v_2,v_3\}) \leq w_e(\{v_1\}) + w_e(\{v_2,v_3\})$$
$$\implies \quad \mathrm{Vol}_{G_e}(\partial\{1,2\}) \geq w_e(\{1,2\}).$$

$$v_1 = 1, \ v_2 = 2, \ v_3 \in \{3,4,5,6\}, \ w_e(\{v_3\}) \geq w_e(\{v_1,v_2,v_3\}) - w_e(\{v_1,v_2\})$$
$$v_1, v_2 \in \{1,2\}, \ v_3 \in \{3,4,5,6\}, \ w_e(\{v_1,v_3\}) \geq w_e(\{v_1\}) + w_e(\{v_1,v_2,v_3\}) - w_e(\{v_1,v_2\})$$
$$v_1, v_2 \in \{1,2\}, \ v_3, v_4, v_5, v_6 \in \{3,4,5,6\}, \ w_e(\{v_1,v_3,v_4\}) \geq w_e(\{v_5,v_6\}) + w_e(\{v_1\}) - w_e(\{v_1,v_2\})$$
$$\implies \quad \mathrm{Vol}_{G_e}(\partial\{1,2\}) \leq 3 w_e(\{1,2\}).$$

**Case $S = \{1,2,3\}$:**

$\mathrm{Vol}_{G_e}(\partial\{1,2,3\})$

$$= -\frac{3}{20}(w_e(\{1\}) + w_e(\{2\}) + w_e(\{3\}) + w_e(\{4\}) + w_e(\{5\}) + w_e(\{6\}))$$

$$+ \frac{5}{12}(w_e(\{1,2\}) + w_e(\{1,3\}) + w_e(\{2,3\} + w_e(\{4,5\}) + w_e(\{4,6\}) + w_e(\{5,6\}))$$

$$- \frac{13}{90}(w_e(\{1,4\}) + w_e(\{1,5\}) + w_e(\{1,6\}) + w_e(\{2,4\}) + w_e(\{2,5\}) + w_e(\{2,6\})$$
$$+ w_e(\{3,4\}) + w_e(\{3,5\}) + w_e(\{3,6\}))$$

$$+ w_e(\{1,2,3\}) + \frac{1}{18}(w_e(\{1,2,4\}) + w_e(\{1,2,5\}) + w_e(\{1,2,6\}) + w_e(\{1,3,4\})$$
$$+ w_e(\{1,3,5\}) + w_e(\{1,3,6\}) + w_e(\{1,4,5\}) + w_e(\{1,4,6\}) + w_e(\{1,5,6\}))$$

$$v_1, v_2, v_3 \in \{1,2,3\}, \ v_4, v_5, v_6 \in \{4,5,6\}, \ w_e(\{v_2\}) + w_e(\{v_3\}) \leq w_e(\{v_1,v_2\}) + w_e(\{v_1,v_3\}),$$
$$w_e(\{v_5\}) + w_e(\{v_6\}) \leq w_e(\{v_4,v_5\}) + w_e(\{v_4,v_6\})$$
$$v_1, v_2, v_3 \in \{1,2,3\}, \ v_4, v_5, v_6 \in \{4,5,6\}, \ w_e(\{v_1,v_2\}) + w_e(\{v_4,v_5\}) \leq w_e(\{v_3,v_6\})$$
$$v_1, v_2, v_3 \in \{1,2,3\}, \ v_4, v_5, v_6 \in \{4,5,6\}, \ w_e(\{v_1,v_2,v_4\}) + w_e(\{v_1,v_4,v_5\}) \leq w_e(\{v_1,v_4\}) + w_e(\{v_3,v_6\})$$
$$\implies \quad \mathrm{Vol}_{G_e}(\partial\{1,2,3\}) \geq w_e(\{1,2,3\}).$$

$v_1, v_2 \in \{1,2,3\}, \ v_4, v_5 \in \{4,5,6\}, \ w_e(\{v_1, v_4, v_5\}) \le w_e(\{v_1, v_4\}) + w_e(\{v_1, v_5\}) - w_e(\{v_1\}),$
$$w_e(\{v_1, v_2, v_4\}) \le w_e(\{v_1, v_4\}) + w_e(\{v_2, v_4\}) - w_e(\{v_4\})$$
$v_1, v_2, v_3 \in \{1,2,3\}, \ v_4, v_5, v_6 \in \{4,5,6\}, \ w_e(\{v_1, v_4\}) \ge w_e(\{v_1\}) + w_e(\{v_5, v_6\}) - w_e(\{v_1, v_2, v_3\})$
$v_1, v_2, v_3 \in \{1,2,3\}, \ v_4, v_5, v_6 \in \{4,5,6\}, \ w_e(\{v_1\}) + w_e(\{v_2\}) \ge w_e(\{v_1, v_2\}),$
$$w_e(\{v_1\}) \ge w_e(\{v_2, v_3\}) - w_e(\{v_1, v_2, v_3\}),$$
$$w_e(\{v_4\}) + w_e(\{v_5\}) \ge w_e(\{v_4, v_5\}),$$
$$w_e(\{v_4\}) \ge w_e(\{v_5, v_6\}) - w_e(\{v_4, v_5, v_6\})$$
$$\implies \quad \mathrm{Vol}_{G_e}(\partial\{1,2,3\}) \le 4 w_e(\{1,2,3\}).$$

**D.5** $\delta(e) = 7, \ \beta^{(e)} = 6$

By using the symmetry property of $w_e(\cdot)$ we have

$w_{12}^{*(e)}$

$= \frac{1}{6}(w_e(\{1\}) + w_e(\{2\})) - \frac{1}{10}(w_e(\{3\}) + w_e(\{4\}) + w_e(\{5\}) + w_e(\{6\}) + w_e(\{7\}))$

$- \frac{1}{6} w_e(\{1,2\}) + \frac{1}{10}(w_e(\{1,3\}) + w_e(\{1,4\}) + w_e(\{1,5\}) + w_e(\{1,6\}) + w_e(\{1,7\})$

$\qquad + w_e(\{2,3\}) + w_e(\{2,4\}) + w_e(\{2,5\}) + w_e(\{2,6\}) + w_e(\{2,7\}))$

$- \frac{1}{12}(w_e(\{3,4\}) + w_e(\{3,5\}) + w_e(\{3,6\}) + w_e(\{3,7\}) + w_e(\{4,5\}) + w_e(\{4,6\})$

$\qquad + w_e(\{4,7\}) + w_e(\{5,6\}) + w_e(\{5,7\}) + w_e(\{6,7\}))$

$- \frac{1}{10}(w_e(\{1,2,3\}) + w_e(\{1,2,4\}) + w_e(\{1,2,5\}) + w_e(\{1,2,6\}) + w_e(\{1,2,7\}))$

$+ \frac{1}{12}(w_e(\{1,3,4\}) + w_e(\{1,3,5\}) + w_e(\{1,3,6\}) + w_e(\{1,3,7\}) + w_e(\{1,4,5\})$

$\qquad + w_e(\{1,4,6\}) + w_e(\{1,4,7\}) + w_e(\{1,5,6\}) + w_e(\{1,5,7\}) + w_e(\{1,6,7\})$

$\qquad + w_e(\{2,3,4\}) + w_e(\{2,3,5\}) + w_e(\{2,3,6\}) + w_e(\{2,3,7\}) + w_e(\{2,4,5\})$

$\qquad + w_e(\{2,4,6\}) + w_e(\{2,4,7\}) + w_e(\{2,5,6\}) + w_e(\{2,5,7\}) + w_e(\{2,6,7\}))$

$- \frac{1}{12}(w_e(\{3,4,5\}) + w_e(\{3,4,6\}) + w_e(\{3,4,7\}) + w_e(\{3,5,6\}) + w_e(\{3,5,7\}) + w_e(\{3,6,7\})$

$\qquad + w_e(\{4,5,6\}) + w_e(\{4,5,7\}) + w_e(\{4,6,7\}) + w_e(\{5,6,7\}))$

**Case $S = \{1\}$:**

$\mathrm{Vol}_{G_e}(\partial\{1\})$

$= w_e(\{1\}) - \frac{1}{3}(w_e(\{2\}) + w_e(\{3\}) + w_e(\{4\}) + w_e(\{5\}) + w_e(\{6\}) + w_e(\{7\}))$

$+ \frac{1}{3}(w_e(\{1,2\}) + w_e(\{1,3\}) + w_e(\{1,4\}) + w_e(\{1,5\}) + w_e(\{1,6\}) + w_e(\{1,7\}))$

$- \frac{2}{15}(w_e(\{2,3\}) + w_e(\{2,4\}) + w_e(\{2,5\}) + w_e(\{2,6\}) + w_e(\{2,7\}) + w_e(\{3,4\})$

$\qquad + w_e(\{3,5\}) + w_e(\{3,6\}) + w_e(\{3,7\}) + w_e(\{4,5\}) + w_e(\{4,6\})$

$\qquad + w_e(\{4,7\}) + w_e(\{5,6\}) + + w_e(\{5,7\}) + w_e(\{6,7\}))$

$+ \frac{2}{15}(w_e(\{1,2,3\}) + w_e(\{1,2,4\}) + w_e(\{1,2,5\}) + w_e(\{1,2,6\}) + w_e(\{1,2,7\}) + w_e(\{1,3,4\})$

$\qquad + w_e(\{1,3,5\}) + w_e(\{1,3,6\}) + w_e(\{1,3,7\}) + w_e(\{1,4,5\}) + w_e(\{1,4,6\})$

$\qquad + w_e(\{1,4,7\}) + w_e(\{1,5,6\})) + w_e(\{1,5,7\}) + + w_e(\{1,6,7\})$

$$v_1 = 1, \ v_2, v_3 \in \{2, 3, 4, 5, 6, 7\}, \ w_e(\{v_1, v_2\}) + w_e(\{v_1, v_3\}) \geq w_e(\{v_2\}) + w_e(\{v_3\})$$
$$v_1 = 1, \ v_2, v_3, v_4, v_5 \in \{2, 3, 4, 5, 6, 7\},$$
$$w_e(\{v_1, v_2, v_3\}) + w_e(\{v_1, v_4, v_5\}) \geq w_e(\{v_2, v_3\}) + w_e(\{v_4, v_5\})$$
$$\implies \mathrm{Vol}_{G_e}(\partial\{1\}) \geq w_e(\{1\}).$$

$$v_1 = 1, \ v_2 \in \{2, 3, 4, 5, 6, 7\}, \ w_e(\{v_1, v_2\}) \leq w_e(\{v_1\}) + w_e(\{v_2\})$$
$$v_1 = 1, \ v_2, v_3 \in \{2, 3, 4, 5, 6, 7\}, \ w_e(\{v_1, v_2, v_3\}) \leq w_e(\{v_1\}) + w_e(\{v_2, v_3\})$$
$$\implies \quad \mathrm{Vol}_{G_e}(\partial\{1\}) \leq 5 w_e(\{1\}).$$

**Case $S = \{1, 2\}$:**

$$\mathrm{Vol}_{G_e}(\partial\{1, 2\})$$
$$= \frac{1}{3}(w_e(\{1\}) + w_e(\{2\})) - \frac{7}{15}(w_e(\{3\}) + w_e(\{4\}) + w_e(\{5\}) + w_e(\{6\}) + w_e(\{7\}))$$
$$+ w_e(\{1, 2\}) - \frac{1}{10}(w_e(\{3, 4\}) + w_e(\{3, 5\}) + w_e(\{3, 6\}) + w_e(\{3, 7\}) + w_e(\{4, 5\})$$
$$+ w_e(\{4, 6\}) + w_e(\{4, 7\}) + w_e(\{5, 6\}) + w_e(\{5, 7\}) + w_e(\{6, 7\}))$$
$$+ \frac{7}{15}(w_e(\{1, 2, 3\}) + w_e(\{1, 2, 4\}) + w_e(\{1, 2, 5\}) + w_e(\{1, 2, 6\} + w_e(\{1, 2, 7\}))$$
$$- \frac{1}{30}(w_e(\{1, 3, 4\}) + w_e(\{1, 3, 5\}) + w_e(\{1, 3, 6\}) + w_e(\{1, 3, 7\}) + w_e(\{1, 4, 5\})$$
$$+ w_e(\{1, 4, 6\}) + w_e(\{1, 4, 7\}) + w_e(\{1, 5, 6\}) + w_e(\{1, 5, 7\}) + w_e(\{1, 6, 7\})$$
$$+ w_e(\{2, 3, 4\}) + w_e(\{2, 3, 5\}) + w_e(\{2, 3, 6\}) + w_e(\{2, 3, 7\}) + w_e(\{2, 4, 5\})$$
$$+ w_e(\{2, 4, 6\}) + w_e(\{2, 4, 7\}) + w_e(\{2, 5, 6\}) + w_e(\{2, 5, 7\}) + w_e(\{2, 6, 7\}))$$
$$+ \frac{1}{6}(w_e(\{3, 4, 5\}) + w_e(\{3, 4, 6\}) + w_e(\{3, 4, 7\}) + w_e(\{3, 5, 6\}) + w_e(\{3, 5, 7\}) + w_e(\{3, 6, 7\})$$
$$+ w_e(\{4, 5, 6\}) + w_e(\{4, 5, 7\}) + w_e(\{4, 6, 7\}) + w_e(\{5, 6, 7\}))$$

$$v_1, v_2 \in \{1, 2\}, \ v_3, v_4, v_5, v_6, v_7 \in \{3, 4, 5, 6, 7\}, \ w_e(\{v_2, v_6, v_7\}) \leq w_e(\{v_1\}) + w_e(\{v_3, v_4, v_5\})$$
$$v_1 = 1, v_2 = 2, v_3, v_4, v_5, v_6, v_7 \in \{3, 4, 5, 6, 7\},$$
$$w_e(\{v_6, v_7\}) \leq w_e(\{v_1, v_2, v_3\}) + w_e(\{v_3, v_4, v_5\}) - w_e(\{v_3\})$$
$$v_1 = 1, v_2 = 2, v_3, v_4 \in \{3, 4, 5, 6, 7\}, \ w_e(\{v_3\}) + w_e(\{v_4\}) \leq w_e(\{v_1, v_2, v_3\}) + w_e(\{v_1, v_2, v_4\})$$
$$\implies \quad \mathrm{Vol}_{G_e}(\partial\{1, 2\}) \geq w_e(\{1, 2\}).$$

$$v_1 = 1, \ v_2 = 2, \ v_3 \in \{3, 4, 5, 6\}, \ w_e(\{v_3\}) \geq w_e(\{v_1, v_2, v_3\}) - w_e(\{v_1, v_2\})$$
$$v_1 = 1, \ v_2 = 2, \ v_3, v_4, v_5, v_6, v_7 \in \{3, 4, 5, 6\}, w_e(\{v_3, v_4\}) \geq w_e(\{v_5, v_6, v_7\}) - w_e(\{v_1, v_2\})$$
$$v_1, \ v_2 \in \{1, 2\}, \ v_3, v_4, v_5, v_6, v_7 \in \{3, 4, 5, 6\},$$
$$w_e(\{v_1, v_3, v_4\}) \geq w_e(\{v_5, v_6, v_7\}) + w_e(\{v_1\}) - w_e(\{v_1, v_2\})$$
$$\implies \quad \mathrm{Vol}_{G_e}(\partial\{1, 2\}) \leq 5 w_e(\{1, 2\}).$$

**Case** $S = \{1, 2, 3\}$:

$\text{Vol}_{G_e}(\partial\{1, 2, 3\})$

$= -\frac{2}{15}(w_e(\{1\}) + w_e(\{2\}) + w_e(\{3\})) - \frac{2}{5}(w_e(\{4\}) + w_e(\{5\}) + w_e(\{6\}) + w_e(\{7\}))$

$+\frac{7}{15}(w_e(\{1, 2\}) + w_e(\{1, 3\}) + w_e(\{2, 3\}))$

$-\frac{1}{6}(w_e(\{1, 4\}) + w_e(\{1, 5\}) + w_e(\{1, 6\}) + w_e(\{1, 7\}) + w_e(\{2, 4\}) + w_e(\{2, 5\})$

$\quad + w_e(\{2, 6\}) + w_e(\{2, 7\}) + w_e(\{3, 4\}) + w_e(\{3, 5\}) + w_e(\{3, 6\}) + w_e(\{3, 7\}))$

$+\frac{1}{10}(w_e(\{4, 5\}) + w_e(\{4, 6\}) + w_e(\{4, 7\}) + w_e(\{5, 6\}) + w_e(\{5, 7\}) + w_e(\{6, 7\}))$

$+w_e(\{1, 2, 3\}) + \frac{2}{15}(w_e(\{1, 2, 4\}) + w_e(\{1, 2, 5\}) + w_e(\{1, 2, 6\}) + w_e(\{1, 2, 7\})$

$\quad + w_e(\{1, 3, 4\}) + w_e(\{1, 3, 5\}) + w_e(\{1, 3, 6\}) + w_e(\{1, 3, 7\})$

$\quad + w_e(\{2, 3, 4\}) + w_e(\{2, 3, 5\}) + w_e(\{2, 3, 6\}) + w_e(\{2, 3, 7\})$

$-\frac{1}{30}(w_e(\{1, 4, 5\}) + w_e(\{1, 4, 6\}) + w_e(\{1, 4, 7\}) + w_e(\{1, 5, 6\}) + w_e(\{1, 5, 7\}) + w_e(\{1, 6, 7\})$

$\quad + w_e(\{2, 4, 5\}) + w_e(\{2, 4, 6\}) + w_e(\{2, 4, 7\}) + w_e(\{2, 5, 6\}) + w_e(\{2, 5, 7\}) + w_e(\{2, 6, 7\})$

$\quad + w_e(\{3, 4, 5\}) + w_e(\{3, 4, 6\}) + w_e(\{3, 4, 7\}) + w_e(\{3, 5, 6\}) + w_e(\{3, 5, 7\}) + w_e(\{3, 6, 7\}))$

$+\frac{1}{2}(w_e(\{4, 5, 6\}) + w_e(\{4, 5, 7\}) + w_e(\{4, 6, 7\}) + w_e(\{5, 6, 7\}))$

$v_1, v_2, v_3 \in \{1, 2, 3\}, \ w_e(\{v_1\}) + w_e(\{v_2\}) \le w_e(\{v_1, v_3\}) + w_e(\{v_2, v_3\}),$
$v_1, v_2, v_3 \in \{1, 2, 3\}, \ v_4, v_5, v_6, v_7 \in \{4, 5, 6, 7\},$
$\qquad\qquad\qquad w_e(\{v_4\}) \le w_e(\{v_1, v_2, v_4\}) + w_e(\{v_4, v_5, v_6\}) - w_e(\{v_3, v_7\}),$
$v_1, v_2, v_3 \in \{1, 2, 3\}, \ v_4, v_5, v_6, v_7 \in \{4, 5, 6, 7\}, \ w_e(\{v_3, v_7\}) \le w_e(\{v_1, v_2\}) + w_e(\{v_4, v_5, v_6\}),$
$v_1, v_2, v_3 \in \{1, 2, 3\}, \ v_4, v_5, v_6, v_7 \in \{4, 5, 6, 7\}, \ w_e(\{v_1, v_4, v_5\}) \le w_e(\{v_2, v_3\}) + w_e(\{v_6.v_7\}),$
$\qquad\qquad\qquad\qquad \implies \quad \text{Vol}_{G_e}(\partial\{1, 2, 3\}) \ge w_e(\{1, 2, 3\}).$

$v_1, v_2, v_3 \in \{1, 2, 3\}, \ v_4, v_5, v_6, v_7 \in \{4, 5, 6, 7\},$
$\qquad\qquad\qquad w_e(\{v_1, v_4, v_5\}) \ge w_e(\{v_2, v_3\}) + w_e(\{v_4, v_5\}) - w_e(\{v_1, v_2, v_3\}),$
$v_1, v_2, v_3 \in \{1, 2, 3\}, \ v_4, v_5, v_6, v_7 \in \{4, 5, 6, 7\},$
$\qquad\qquad\qquad w_e(\{v_1, v_4\}) \ge w_e(\{v_1\}) + w_e(\{v_5, v_6, v_7\}) - w_e(\{v_1, v_2, v_3\}),$
$v_1, v_2 \in \{1, 2, 3\}, \ v_3 \in \{4, 5, 6, 7\}, \ w_e(\{v_3\}) \ge w_e(\{v_1, v_2, v_3\}) - w_e(\{v_1, v_2\}),$
$v_1, v_2, v_3 \in \{1, 2, 3\}, \ w_e(\{v_1\}) \ge w_e(\{v_2, v_3\}) - w_e(\{v_1, v_2, v_3\}),$
$\qquad\qquad\qquad\qquad \implies \quad \text{Vol}_{G_e}(\partial\{1, 2, 3\}) \le 6w_e(\{1, 2, 3\}).$

# E  Proof of Theorem 3.4

Suppose that $\{f_{v\tilde{v}}^o\}_{\{v\tilde{v} \in E^{(e)}\}}$ and $\beta^o$ represent the optimal solution of the optimization problem (11). To prove that the values of $\beta^o$ are equal to those listed in Table 1, we proceed as follows. The result of the optimization procedure over $\{f_{v\tilde{v}}\}_{\{v\tilde{v} \in E^{(e)}\}}$ produces the weights (coefficients) of the linear mapping. The optimization problem (11) may be rewritten as

$$\min_{\{f_{v\tilde{v}}\}_{\{v,\tilde{v}\} \in E_o}} \quad \beta \tag{20}$$

$$\text{s.t.} \quad w_e(S) \le \sum_{v \in S, \tilde{v} \in e/S} f_{v\tilde{v}}(w_e) \le \beta w_e(S), \quad \forall \, S \in 2^e \text{ and submodular } w_e(\cdot).$$

which is essential a linear programming. However, as there are uncountable many choices for the submodular functions $w_e(\cdot)$, the above optimization problem has uncountable many constraints.

However, given a finite collection of inhomogenous cost functions $\Omega = \{w_e^{(1)}(\cdot), w_e^{(2)}(\cdot), ...\}$ all of which are submodular, the following linear program

$$\min_{\{f_{v\tilde{v}}\}_{\{v,\tilde{v}\}\in E_o}} \quad \beta \tag{21}$$

$$\text{s.t.} \quad w_e^{(r)}(S) \leq \sum_{v\in S, \tilde{v}\in e/S} f_{v\tilde{v}}(w_e^{(r)}) \leq \beta w_e^{(r)}(S), \quad \text{for all } S \in 2^e \text{ and } w_e^{(r)}(\cdot) \in \Omega.$$

can be efficiently computed and yields an optimal $\beta^\Omega$ that provides a lower bound for $\beta^o$. Therefore, we just need to identify the sets $\Omega$ for different values of $\delta(e)$ that meet the values of $\beta^{(e)}$ listed in Table 1.

The proof involves solving the linear program (21). We start by identifying some structural properties of the problem.

**Proposition E.1.** Given $\delta(e)$, the optimization problem (11) over $\{f_{v\tilde{v}}\}_{\{v\tilde{v}\in E^{(e)}\}}$ involves $3[\frac{\delta(e)}{2}] - 1$ variables, where $[a]$ denotes the largest integer not greater than $a$.

*Proof.* The linear mapping $f_{v\tilde{v}}$ may be written as

$$f_{v\tilde{v}}(w_e) = \sum_{S\in 2^e} \phi(v\tilde{v}, S) w_e(S),$$

where $\phi(v\tilde{v}, S)$ represent the coefficients that we wish to optimize, and which depend on the edge $v\tilde{v}$ and the subset $S$. Although we have $\binom{\delta(e)}{2} \times 2^{\delta(e)}$ coefficients, the coefficients are not independent from each other. To see what kind of dependencies exist, define the set of all permutations of the vertices of $e$ $\pi : e \to \{1, 2, ..., \delta(e)\}$; clearly, there are $\delta(e)!$ such permutations $\pi$. Also, define $\pi(S) = \{\pi(v)|v \in S\}$ for $S \subseteq e$. If a set function $w(\cdot)$ over all the subsets of $\{1, 2, ..., \delta(e)\}$ satisfies the following conditions

$$\begin{aligned} w(\emptyset) &= 0, \\ w(S) &= w(\bar{S}), \\ w(S_1) + w(S_2) &\geq w(S_1 \cap S_2) + w(S_1 \cup S_2), \end{aligned}$$

for $S, S_1, S_2 \subseteq \{1, 2, ..., \delta(e)\}$, then one may construct $\delta(e)!$ many inhomogeneous cost functions $w_e^{(\pi)}(\cdot)$ such that for all distinct $\pi$ one has $w_e^{(\pi)}(\cdot) = w(\pi(\cdot))$. As all the weights $w_e^{(\pi)}$ are submodular and appear in the constraints of the optimization problem (20), the coefficients $\phi(v\tilde{v}, S)$ will be invariant under the permutations $\pi$; thus, they will depend only on two parameters, $|\{v, \tilde{v}\} \cap S|$ and $|S|$. We replace $\phi$ with another function $\tilde{\phi}$ to capture this dependence

$$\phi(v\tilde{v}, S) = \phi(\pi(v)\pi(\tilde{v}), \ \pi(S)) = \tilde{\phi}(|\{v, \tilde{v}\} \cap S|, |S|).$$

Moreover, as $w_e(S)$ is symmetric, i.e., as $w_e(S) = w_e(\bar{S})$, we also have

$$\tilde{\phi}(|\{v, \tilde{v}\} \cap \bar{S}|, |\bar{S}|) = \tilde{\phi}(|\{v, \tilde{v}\} \cap S|, |S|).$$

Hence, for any given $\delta(e)$, the set of the coefficients of the linear function may be written as

$$\Phi = \{\tilde{\phi}(r, s)|(r, s) \in \{0, 1, 2\} \times \{1, 2, 3, ..., \delta(e) - 2, \delta(e) - 1\}/\{(2, 1), (0, \delta(e) - 1)\},$$

$$\text{s.t.} \quad \tilde{\phi}(0, s) = \tilde{\phi}(2, \delta(e) - s), \ \tilde{\phi}(1, s) = \tilde{\phi}(1, \delta(e) - s)\}.$$

which concludes the proof. $\square$

Using Proposition E.1, we can transform the optimization problem (21) into the following form:

$$\min_{\Phi} \quad \beta$$

$$\text{s.t.} \ w_e^{(r)}(S) \leq \sum_{v\in S, \tilde{v}\in e/S} \sum_{S'\subseteq e} \tilde{\phi}(|\{v, \tilde{v}\} \cap S'|, |S'|) w_e^{(r)}(S') \leq \beta w_e^{(r)}(S), \ \forall S \in 2^e, \forall w_e^{(r)}(\cdot) \in \Omega.$$

For a given $\delta(e)$, we list the sets $\Omega = \{w_e^{(1)}, w_e^{(2)}, ...\}$ in Table. 6. The above linear program yields optimal values of $\beta^\Omega$ equal to those listed in Table 1. As already pointed out, the cases $\delta(e) = 2, 3$ are simple to verify, and hence we concentrate on the sets $\Omega$ for $\delta(e) \geq 4$. The case $\delta(e) = 7$ is handled similarly but requires a large verification table that we omitted for succinctness.

## F  Proof of Theorem 3.5

For two pairs of vertices $v\tilde{v}, u\tilde{u} \in E^{(e)}$, count the number of $S \in 2^e/\{\emptyset, e\}$'s with $|S| = k$ and $u\tilde{u} \in \partial S$ that satisfy the following conditions:

1) $v, \tilde{v} \in \{u, \tilde{u}\}$ and $v \in S, \tilde{v} \in \bar{S}$: The number is $\binom{\delta(e)-2}{k-1}$;

2) $v, \tilde{v} \in \{u, \tilde{u}\}$ and $v, \tilde{v} \in \bar{S}$: The number is 0;

3) $v \in \{u, \tilde{u}\}, \tilde{v} \notin \{u, \tilde{u}\}$ and $v \in S, \tilde{v} \in \bar{S}$: The number is $\binom{\delta(e)-3}{k-1}$;

4) $v \in \{u, \tilde{u}\}, \tilde{v} \notin \{u, \tilde{u}\}$ and $v, \tilde{v} \in \bar{S}$: The number is $\binom{\delta(e)-3}{k-1}$;

5) $v \notin \{u, \tilde{u}\}, \tilde{v} \in \{u, \tilde{u}\}$ and $v \in S, \tilde{v} \in \bar{S}$: The number is $\binom{\delta(e)-3}{k-2}$;

6) $v \notin \{u, \tilde{u}\}, \tilde{v} \in \{u, \tilde{u}\}$ and $v, \tilde{v} \in \bar{S}$: The number is $\binom{\delta(e)-3}{k-1}$;

7) $v, \tilde{v} \notin \{u, \tilde{u}\}$ and $v \in S, \tilde{v} \in \bar{S}$: The number is $2\binom{\delta(e)-4}{k-2}$;

8) $v, \tilde{v} \notin \{u, \tilde{u}\}$ and $v, \tilde{v} \in \bar{S}$: The number is $2\binom{\delta(e)-4}{k-1}$.

Moreover, some identities in the following can be demonstrated:

$$\sum_{k=1}^{\delta(e)-1} \binom{\delta(e)-3}{k-1} \frac{1}{k(\delta(e)-k)} \overset{a)}{=} \sum_{k=2}^{\delta(e)-1} \binom{\delta(e)-3}{k-2} \frac{1}{k(\delta(e)-k)} \overset{b)}{=} \sum_{k=1}^{\delta(e)-2} \binom{\delta(e)-3}{k-1} \frac{1}{(k+1)(\delta(e)-k-1)};$$

$$\sum_{k=1}^{\delta(e)-1} \binom{\delta(e)-4}{k-2} \frac{1}{k(\delta(e)-k)} \overset{c)}{=} \sum_{k=1}^{\delta(e)-2} \binom{\delta(e)-4}{k-1} \frac{1}{(k+1)(\delta(e)-k-1)}.$$

where the equalities are by substitution: a) $k \to \delta(e) - k$, b) $k \to k + 1$, c) $k \to \delta(e) - (k + 1)$.

As we assume that $w_e(S) = \sum_{u\tilde{u} \in \partial S} w_{u\tilde{u}}^{(e)}$, the RHS of formula (10) can be decomposed into the weighted sum of $w_{u\tilde{u}}^e$. As $w_e$ is symmetric and the above identities can be used,

$$w_{v\tilde{v}}^{*(e)} = \sum_{S \in 2^e/\{\emptyset, e\}} \left[ \frac{w_e(S)}{|S|(\delta(e)-|S|)} 1_{v \in S, \tilde{v} \in \bar{S}} - \frac{w_e(S)}{(|S|+1)(\delta(e)-|S|-1)} 1_{v, \tilde{v} \in \bar{S}} \right]$$

$$= \sum_{k=1}^{\delta(e)-1} \sum_{S:|S|=k} \frac{w_e(S)}{k(\delta(e)-k)} 1_{v \in S, \tilde{v} \in \bar{S}} - \sum_{k=1}^{\delta(e)-2} \sum_{S:|S|=k} \frac{w_e(S)}{(k+1)(\delta(e)-k-1)} 1_{v, \tilde{v} \in \bar{S}}$$

$$= \sum_{k=1}^{\delta(e)-1} \sum_{S:|S|=k} \frac{1}{k(\delta(e)-k)} 1_{v \in S, \tilde{v} \in \bar{S}} \sum_{u\tilde{u} \in \partial S} w_{u\tilde{u}}^e - \sum_{k=1}^{\delta(e)-2} \sum_{S:|S|=k} \frac{1}{(k+1)(\delta(e)-k-1)} 1_{v, \tilde{v} \in \bar{S}} \sum_{u\tilde{u} \in \partial S} w_{u\tilde{u}}^e$$

$$= \sum_{u\tilde{u} \in \partial S} w_{u\tilde{u}}^{(e)} \left\{ \sum_{k=1}^{\delta(e)-1} \binom{\delta(e)-2}{k-1} \frac{1}{k(\delta(e)-k)} 1_{v, \tilde{v} \in \{u, \tilde{u}\}} \right.$$

$$+ \left[ \sum_{k=1}^{\delta(e)-1} \binom{\delta(e)-3}{k-1} \frac{1}{k(\delta(e)-k)} - \sum_{k=1}^{\delta(e)-2} \binom{\delta(e)-3}{k-1} \frac{1}{(k+1)(\delta(e)-k-1)} \right] 1_{v \in \{u, \tilde{u}\}, \tilde{v} \notin \{u, \tilde{u}\}}$$

$$+ \left[ \sum_{k=1}^{\delta(e)-1} \binom{\delta(e)-3}{k-2} \frac{1}{k(\delta(e)-k)} - \sum_{k=1}^{\delta(e)-2} \binom{\delta(e)-3}{k-1} \frac{1}{(k+1)(\delta(e)-k-1)} \right] 1_{v \notin \{u, \tilde{u}\}, \tilde{v} \in \{u, \tilde{u}\}}$$

$$+ 2 \left[ \sum_{k=1}^{\delta(e)-1} \binom{\delta(e)-4}{k-2} \frac{1}{k(\delta(e)-k)} - \sum_{k=1}^{\delta(e)-2} \binom{\delta(e)-4}{k-1} \frac{1}{(k+1)(\delta(e)-k-1)} \right] 1_{v \notin \{u, \tilde{u}\}, \tilde{v} \notin \{u, \tilde{u}\}} \right\}$$

$$= \sum_{k=1}^{\delta(e)-1} \binom{\delta(e)-2}{k-1} \frac{1}{k(\delta(e)-k)} w_{v\tilde{v}}^{(e)} = \frac{2^{\delta(e)}-2}{\delta(e)(\delta(e)-1)} w_{v\tilde{v}}^{(e)}$$

which concludes the proof.

# G Complexity analysis

Recall that the proposed algorithm consists of three computational steps: 1) Projecting each InH-hyperedge onto a subgraph; 2) Combining the subgraphs to create a graph; 3) Performing spectral clustering on the derived graph based on Algorithm 1 (Appendix. B). The complexity of the algorithms depends on the complexity of these three steps. Let $\delta^* = \max_{e \in E} \delta(e)$ denote the largest size of a hyperedge. If in the first step we solve the optimization procedure (5) for all InH-hyperedges with at most $2^{\delta(e)}$ constraints, the worst case complexity of the algorithm is $O(2^{c\delta^*}|E|)$, where $c$ is a constant that depends on the LP-solver. The second step has complexity $O((\delta^*)^2|E|)$, while the third step has complexity $O(n^2)$, given that one has to find the eigenvectors corresponding to the extremal eigenvalues. Other benchmark hypergraph clustering algorithms, such as Clique Expansion, Star Expansion [14] and Zhou's normalized hypergraph cut [11] share two steps of our procedure and hence have the same complexity for the corresponding computations. In practice, we usually deal with hyperedges of small size ($< 10$) and hence $\delta^*$ may, for all purposes, be treated as a constant. Hence, the complexity overhead of our method is of the same order as that of the last two steps, and hence we retain the same order of computation as classical homogeneous clustering methods. Nevertheless, to reduce the complexity of InH-partition, one may use predetermined mappings of the form (9) and (10). In the applications discussed in what follows, we exclusively used this computationally efficient approach.

# H Discussion of Equation (9)

For the case that only the values of $w_e(\{v\})$ are known, we showed that one may perform perfect projections with $\beta^{(e)} = 1$. Suppose now that $w^{(e)}$ takes the form (9), i.e., that for each pair of vertices $v\tilde{v}$ in $e$, one has

$$w_{v\tilde{v}}^{(e)} = \frac{1}{\delta(e) - 2}\left[w_e(\{v\}) + w_e(\{\tilde{v}\})\right] - \frac{1}{(\delta(e) - 1)(\delta(e) - 2)}\sum_{v' \in e} w_e(\{v'\}).$$

For all $v \in e$, one can compute

$$\sum_{\tilde{v} \in e/\{v\}} w_{v\tilde{v}}^{(e)} = \frac{\delta(e) - 1}{\delta(e) - 2}w_e(\{v\}) + \frac{1}{\delta(e) - 2}\sum_{\tilde{v} \in e/\{v\}} w_e(\{\tilde{v}\}) - \frac{1}{\delta(e) - 2}\sum_{v' \in e} w_e(\{v'\})$$

$$= w_e(\{v\},$$

which confirms that the projection is perfect. When $\delta(e) > 3$, a perfect projection is not unique as the problem is underdetermined. It is also easy to check the conditions for nonnegativity of the components of $w^{(e)}$: For each pair of vertices $v\tilde{v}$ in $e$, one requires

$$w_e(\{v\}) + w_e(\{\tilde{v}\}) \geq \frac{1}{(\delta(e) - 1)}\sum_{v' \in e} w_e(\{v'\}), \tag{22}$$

which indicates the weights in $w_e(\cdot)$ associated with different vertices should satisfy a special form of a balancing condition.

# I Supplementary Applications and Experiments

## I.1 Structure Learning of Ranking Data

**Synthetic data.** We first compare the InH-partition (InH-Par) method with the AnchorsPartition (APar) technique proposed in [34] on synthetic data. Note that APar is assumed to know the correct size of the riffled-independent sets while InH-partition automatically determines the sizes of the parts. We set the number elements to $n = 16$, and partitioned them into a pair $(S^*, \bar{S}^*)$, where $|S^*| = q$, $1 \leq q \leq n$. For a sample set size $m$, we first independently choose scores $s_i, \bar{s}_i \sim$Uniform([0,1]). For $i \in V$ and then generate $m$ rankings via the following procedure: We first use the Plackett-Luce Model [40] with parameters $s_i, i \in S^*$ and $\bar{s}_i, i \in \bar{S}^*$, to generate $\sigma_{S^*}$ and $\sigma_{\bar{S}^*}$. Then, we interleave $\sigma_{S^*}$ and $\sigma_{\bar{S}^*}$, which were sampled uniformly at random without replacement, to form $\sigma$. The performance of the method is characterized via the success rate of full recovery of $(S^*, \bar{S}^*)$.

Figure 4: Success rate vs Sample Complexity & Triple-sampling Rate. a),c): $q = 4$ with scores $s_i$; b),d): $q = 8$ with scores $s_i$; e) $q = 4$ with scores $s_i^3$; f): $q = 8$ with scores $s_i^3$.

The results of various algorithms based on 100 independently generated sample sets are listed in Figure 4 a) and b). For almost all $m$, InH-partition outperforms APar. Only when $q = 4$ and the sample size $m$ is large, InH-partition may offer worse performance than Apar. The explanation for this finding is that InH-partition performs a normalized cut that tends to balance the sizes of different classes. With regards to the computational complexity of the methods, both require one to evaluate the mutual information of all triples of elements at the cost of $O(mn^3)$ operations. To reduce the time complexity of this step, one may sample each triple independently with probability $r$. Results pertaining to triple-sampling with $m = 10^4$ are summarized in Figure 4 c) and d). The InH-partition can achieve high success rate 80% even when only a small fraction of triples ($r < 0.2$) is available. On the other hand, APar only works when almost all triples are sampled ($r > 0.7$).

To further test the performance of InH-partition, instead of using the previously described $s_i$ values as the parameters for Plakett-Luce Model, we use the values $s_i^3$ instead. This choice of parameters further restricts the positions of the candidates within $S^*$ and $\bar{S}^*$. Hence, the mutual information of interest is closer to zero and hence harder to estimate. The results for this setting are shown in part e) and f) of Figure 4. As may be seen, in this setting, the performance of APar is poor while that of InH-partition changes little.

**Real data - Supplement for the Irish Election Dataset [38].** The Irish Election Dataset consists of rankings of 14 candidates from different parties, listed in Table 2. This information was used to perform the learning tasks described in the main text.

Table 2: List of candidates from the Meath Constituency Election in 2002 (reproduced from [34, 38])

| | Candidate | Party | | Candidate | Party |
|---|---|---|---|---|---|
| 1 | Brady, J. | Fianna Fáil | 8 | Kelly, T. | Independent |
| 2 | Bruton, J. | Fine Gael | 9 | O'Brien, P. | Independent |
| 3 | Colwell, J. | Independent | 10 | O'Byrne, F. | Green Party |
| 4 | Dempsey, N. | Fianna Fáil | 11 | Redmond, M. | Christian Solidarity |
| 5 | English, D. | Fine Gael | 12 | Reilly, J. | Sinn Féin |
| 6 | Farrelly, J. | Fine Gael | 13 | Wallace, M. | Fianna Fáil |
| 7 | Fitzgerald, B. | Independent | 14 | Ward, P. | Labour |

Table 3: List of 10 sushi from the sushi preference dataset (reproduced from [41])

| | Sushi | Type | | Candidate | Party |
|---|---|---|---|---|---|
| 1 | ebi | shrimp | 6 | sake | salmon roe |
| 2 | anago | sea eel | 7 | tamago | egg |
| 3 | maguro | tuna | 8 | toro | fatty tuna |
| 4 | ika | squid | 9 | tekka-maki | tuna roll |
| 5 | uni | sea urchin | 10 | kappa-maki | cucumber roll |

Figure 5: Hierarchical partitioning structure of sushi preference detected by InH-Par

In addition, we performed the same structure learning task on the sushi preference ranking dataset [41]. This dataset consists of 5000 full rankings of ten types of sushi. The different types of sushi evaluated are listed in Table 3. We ran InH-partition to split the ten sushi types to obtain a hierarchical clustering structure as the one shown in Figure 5. The figure reveals two meaningful clusters, $\{5, 6\}$ (uni,sake) and $\{3, 8, 9\}$ (tuna-related sushi): The sushi types labeled by 5 and 6 have the commonality of being expensive and branded as "daring, luxury sushi," while sushi types labeled by $3, 8, 9$ all contain tuna. InH-partition cannot detect the so-called "vegeterian-choice sushi" cluster $\{7, 10\}$, which was recovered by Apar [34]. This may be a consequence of the ambiguity and overlap of clusters, as the cluster $\{4, 7\}$ may also be categorized as "rich in lecithin". The detailed comparisons between InH-partition and APar are performed based on their ability to detect the two previously described standard clusters, $\{5, 6\}$ and $\{3, 8, 9\}$, using small training sets. The averaged results based on 100 independent tests are depicted in Figure 6 a). As may be seen, InH-partition outperforms APar in recovering both the clusters (uni,sake) and (tuna sushi), and hence is superior to APar when learning both classes simultaneously. We also compared InH-partition and APar in the large sample regime ($m = 5000$) while using only a subset of triples. The averaged results over 100 sets of independent samples are shown Figure 6 b), again indicating the robustness of InH-partition to missing triple information.

### I.2 Subspace segmentation

Subspace segmentation is an extension of traditional data segmentation problems that has the goal to partition data according to their intrinsically embedded subspaces. Among subspace segmentation methods, those based on hypergraph clustering exhibit superior performance compared to others [42]. They also exhibit other distinguishing features, such as loose dependence on the choice of parameters [14], robustness to outliers [3, 5], and clustering robustness and accuracy [43].

Hypergraph clustering algorithms are exclusively homogeneous: If the intrinsic affine space is $p$-dimensional ($p$-D), the algorithms use $\psi$-uniform ($\psi > p + 1$, typically set to $p + 2$) hypergraphs $\mathcal{H} = (V, E)$, where the vertices in $V$ correspond to observed data vectors and the hyperedges in $E$ are chosen $\psi$-tuples of vertices. To each hyperedge $e$ in the hypergraph $\mathcal{H}$ one assigns a weight $w_e$, typically of the form $w_e = \exp(-d_e^2/\theta^2)$, where $d_e$ describes the deviation needed to fit the corresponding $\psi$-tuple of vectors into a $p$-D affine subspace, and $\theta$ represents a tunable parameter obtained by cross validation [14] or computed empirically [43]. A small value of $d_e$ corresponds to a large value of $w_e$, and indicates that $\psi$-tuples of vectors in $e$ tend to be clustered together. As a good fit of the subspace yields a large weight for the corresponding hyperedge, hypergraph clustering tends to avoid cutting hyperedges of large weight and thus mostly groups vectors within one subspace together. The performance of the methods varies due to different techniques used for computing the deviation $d_e$ and for sampling the hyperedges. Some widely used deviations include $d_e^{\text{H}-1}$, defined as the mean Euclidean distance to the optimal fitted affine subspace [14, 3, 5, 4], and the *polar*

Figure 6: Clusters detected in the sushi preference dataset: a) Success rate vs Sample Complexity; b) Success rate vs Triple-Sampling Probability.

*curvature* (PC) [43], both of which lead to a homogeneous partition. Instead, we propose to use an inhomogeneous deviation defined as

$$d_e^{\text{InH}}(\{v\}) = \text{Euclidean distance between } v \text{ and the affine subspace generated by } e/\{v\}, \text{ for all } v \in e.$$

This deviation measures the "distance" needed to fit $v$ into the subspace supported by $e/v$ and will be used to construct inhomogeneous cost functions $w_e(\cdot)$ via $w_e(\{v\}) = \exp[-d_e^{\text{InH}}(\{v\})^2/\theta^2]$, as described in what follows. Note that the choice of a "good" deviation is still an open problem, which may depend on specific datasets. Hence, to make a comprehensive comparison, besides $d_e^{\text{H}-1}$ and PC, we also made use of another homogeneous deviation, $d_e^{\text{H}-2} = \sum_{v \in e} d_e^{\text{InH}}(\{v\})/\delta(e)$ which is the average of all the defined inhomogeneous deviations. Comparing the results obtained from $d_e^{\text{InH}}$ with $d_e^{\text{H}-2}$ will highlight the improvements obtained from InH-partition, rather than from the choice of the deviation. The inhomogeneous form of deviation $d_e^{\text{InH}}(\cdot)$ has a geometric interpretation based on the polytopes ($p = 1$) shown in Figure 7 (with $d_e^{\text{H}-1}$ and $d_e^{\text{H}-2}$). There, $d_e^{\text{InH}}(\{v\})$ is the distance of $\{v\}$ from the hyperplane spanned by $e/\{v\}$. The induced inhomogeneous weight $w_e(\{v\}) = \exp(-d_e^{\text{InH}}(\{v\})^2/\theta^2)$ may be interpreted as the cost of separating $\{v\}$ away from the other points (vertices) in $e$.

$$d_e^{\text{H}-1} = (\ell_i + \ell_j + \ell_k)/3$$
$$d_e^{\text{H}-2} = (h_i + h_j + h_k)/3$$
$$d_e^{\text{InH}}(\{v_i\}) = h_i$$

Figure 7: Illustration of the deviation ($p = 1$) used for subspace segmentation.

All hypergraph-partitioning based subspace segmentation algorithms essentially use the NCut procedure described in the main text, but their performances vary due to different approaches for constructing the hypergraphs. Three steps in the clustering procedure are key to the performance quality: The first is to quantify the deviation to fit a collection of vectors into a affine subspace; the second is to choose the parameter $\theta$; the third is to sample $\psi$-tuples of vectors, i.e., choose the hyperedges of the hypergraph. For fairness of comparison, in all our experiments we computed an inhomogeneous deviation $d_e$ for the hyperedge $e$ instead of a homogeneous one in the first step, and kept the other two key steps the same as used in the standard literature. In particular, we performed hyperedge sampling uniformly at random for the experiments pertaining to $k$-line segmentation; we used the same hyperedge sampling procedure as that of SCC [43] for the experiments pertaining to motion segmentation. The reason for these two different types of settings are to assess the contribution of InH methods, rather than the sampling procedure.

Table 4: The directions of the $k$-lines.

| k =2 | k =3 | k =4 |
|---|---|---|
| (0.97,0.26,0.00)<br>(0.97,-0.26,0.00) | (0.95,0.30,0.00)<br>(0.95,-0.15,0.26)<br>(0.95,-0.15,-0.26) | (0.93,0.37,0.00)<br>(0.93,0.00,0.37)<br>(0.93,-0.37,0.00)<br>(0.93,0.00,-0.37) |

Figure 8: Misclassification rate (mean and standard deviation) vs noise level: a) $k = 2$; b) $k = 3$; c) $k = 4$.

Our first experiment pertains to segmenting $k$-lines in a 3D Euclidean space ($D = 3, p = 1, k = 2, 3, 4$). The $k$-lines all pass through the origin, and their directions, listed in Table 4, are such that the minimal angles between two lines are restricted to 30 degree; 40 points are sampled uniformly from the segment of each line lying in the unit ball so there are $40k$ points in total. Each point is independently corrupted by 3D mean-zero Gaussian noise with covariance matrix $\theta_n^2 \mathbf{I}$. We determined the parameter $\theta$ through cross validation and uniformly at random picked $100 \times k^2$ many triples. We computed the percentage of misclassified points based on 50 independent tests; the misclassification rate is denoted by $e\%$ and the results are shown in Figure 8. The InH-partition only has $50\%$ of the misclassification errors of H-partition, provided that the noise is small ($\theta_n < 0.01$). To see why this may be the case, let us consider a triple of datapoints $\{v_i, v_j, v_k\}$ where $v_i$ and $v_j$ belong to the same cluster, while $v_k$ may belong to a different cluster. The line that goes through $v_i$ and $v_j$ is close to the true affine subspace when the noise is small and thus the distance from the third point $v_k$ to this line can serve as a precise indicator whether $v_k$ lies within the same true affine subspace. When the noise is high, the InH-partition also performs better when the number of classes is $k = 2$, but starts to deteriorate in performance as $k$ increases. The reason behind this phenomena is as follows: Inhomogenous costs of a hyperedge provide more accurate information about the subspaces than the homogenous costs when at least two points of the hyperedge belong to the same line cluster. This is due to the definition of the deviation $d_e^{\text{inH}}$; but hyperedges of this type become less likely as $k$ increases.

The second problem we investigated in the context of subspace clustering is motion segmentation. Motion segmentation, a widely used application in computer vision, is the task of clustering point trajectories extracted from a video of a scene according to different rigid-body motions. The problem can be reduced to a subspace clustering problem as all the trajectories associated with one motion lie in one specified 3D affine subspace ($p = 3$) [44]. We evaluate the performance of the InH-partition method over the well-known motion segmentation dataset, Hopkins155 [45]. This dataset consists of 155 sequences of two and three motions from three categories of scenes: Checkerboard, traffic and articulated sequences. Our experiments show the InH-partition algorithm outperforms the benchmark algorithms based on the use of H-partitions over this dataset including spectral curvature clustering technique (SCC [43]). To make the comparison fair, we simply replaced the homogeneous distance *polar curvature* in SCC with the inhomogeneous distance $d_e^{\text{InH}}$, the homogenous distances $d_e^{\text{H}-1}$ and $d_e^{\text{H}-2}$, and keep all other steps the same. We also evaluated the performance of some other methods, including Generalized PCA (GPCA) [46], Local Subspace Affinity [47], Agglomerative Lossy Compression (ALC) [48], and Sparse Subspace Clustering (SSC) [49]. The results based on the average over 50 runs for each video are shown in Table 5.

As may be seen, InH-partition outperforms all methods except for SSC (not based on hypergraph clustering), which shows the superiority of replacing H-hyperedges with inhomogeneous ones. Although InH-partition fails to outperform SSC, it has significantly lower complexity and is much easier to use and implement in practice. In addition, some recent algorithms based on H-partitions may leverage the complex hyperedge-sampling steps for this application [50], and we believe that the InH-partition method can be further improved by changing the sampling procedure, and made more appropriate for inhomogeneous hypergraph clustering as opposed to SSC. This topic will be addressed elsewhere.

Table 5: Misclassification rates $e\%$ for the Hopkins 155 dataset. (MN: mean; MD: median)

| Method | Two Motions | | | | | | | | Three Motions | | | | | | | |
|---|---|---|---|---|---|---|---|---|---|---|---|---|---|---|---|---|
| | Chck.(78) | | Trfc.(31) | | Artc.(11) | | All(120) | | Chck.(26) | | Trfc.(7) | | Artc.(2) | | All(115) | |
| | MN | MD | MN | MD | MN | MD | MN | MD | MN | MD | MN | MD | MN | MD | MN | MD |
| GPCA [46] | 6.09 | 1.03 | 1.41 | 0.00 | 2.88 | 0.00 | 4.59 | 0.38 | 31.95 | 32.93 | 19.83 | 19.55 | 16.85 | 16.85 | 28.66 | 28.26 |
| LSA [47] | 2.57 | 0.27 | 5.43 | 1.48 | 4.10 | 1.22 | 3.45 | 0.59 | 5.80 | 1.77 | 25.07 | 23.79 | 7.25 | 7.25 | 9.73 | 2.33 |
| ALC [48] | 1.49 | 0.27 | 1.75 | 1.51 | 10.70 | 0.95 | 2.40 | 0.43 | 5.00 | 0.66 | 8.86 | 0.51 | 21.08 | 21.08 | 6.69 | 0.67 |
| SSC [49] | 1.12 | 0.00 | 0.02 | 0.00 | 0.62 | 0.00 | 0.82 | 0.00 | 2.97 | 0.27 | 0.58 | 0.00 | 1.42 | 1.42 | 2.45 | 0.20 |
| SCC [43] | 1.77 | 0.00 | 0.63 | 0.14 | 4.02 | 2.13 | 1.68 | 0.07 | 6.23 | 1.70 | 1.11 | 1.40 | 5.41 | 5.41 | 5.16 | 1.58 |
| H+$d_e^{H-1}$ | 12.27 | 5.06 | 14.91 | 9.94 | 12.85 | 3.66 | 12.92 | 6.01 | 22.13 | 23.98 | 21.99 | 18.12 | 19.79 | 19.79 | 21.97 | 20.45 |
| H+$d_e^{H-2}$ | 4.20 | 0.43 | 0.33 | 0.00 | 1.53 | 0.10 | 2.93 | 0.06 | 7.05 | 2.22 | 7.02 | 3.98 | 6.47 | 6.47 | 7.01 | 2.12 |
| InH-par | 1.69 | 0.00 | 0.61 | 0.22 | 1.22 | 0.62 | 1.40 | 0.04 | 4.82 | 0.69 | 2.46 | 0.60 | 4.23 | 4.23 | 4.06 | 0.65 |

# J Supplementary Tables

Table 6: Inhomogeous cost functions $w_e^{(r)}(S)$ for $\delta(e) \in \{4, 5, 6\}$.

$\delta(e) = 4,\ \beta^{(e)} = 3/2$

| r | 1 | 2 | 3 | 4 | 1, 2 | 1, 3 | 1, 4 |
|---|---|---|---|---|------|------|------|
| 1 | 0 | 1 | 1 | 1 | 1 | 1 | 1 |
| 2 | 0 | 0 | 1 | 1 | 0 | 1 | 1 |
| 3 | 1 | 1 | 1 | 1 | 1 | 1 | 1 |
| 4 | 1 | 1 | 1 | 1 | 2 | 2 | 2 |

$\delta(e) = 5,\ \beta^{(e)} = 2$

| r | 1 | 2 | 3 | 4 | 5 | 1, 2 | 1, 3 | 1, 4 | 1, 5 | 2, 3 | 2, 4 | 2, 5 | 3, 4 | 3, 5 | 4, 5 |
|---|---|---|---|---|---|------|------|------|------|------|------|------|------|------|------|
| 1 | 0 | 1 | 1 | 1 | 1 | 1 | 1 | 1 | 1 | 1 | 1 | 1 | 1 | 1 | 1 |
| 2 | 0 | 1 | 1 | 1 | 1 | 1 | 1 | 1 | 1 | 2 | 2 | 2 | 2 | 2 | 2 |
| 3 | 1 | 1 | 1 | 1 | 1 | 1 | 1 | 1 | 1 | 1 | 1 | 1 | 1 | 1 | 1 |
| 4 | 1 | 1 | 1 | 1 | 1 | 1 | 2 | 2 | 2 | 2 | 2 | 2 | 1 | 1 | 1 |
| 5 | 1 | 1 | 1 | 1 | 1 | 0 | 2 | 2 | 2 | 2 | 2 | 2 | 1 | 1 | 1 |
| 6 | 1 | 1 | 1 | 1 | 1 | 2 | 2 | 2 | 2 | 2 | 2 | 2 | 2 | 2 | 2 |

$\delta(e) = 6,\ \beta^{(e)} = 4$

| r | 1 | 2 | 3 | 4 | 5 | 6 | 1, 2 | 1, 3 | 1, 4 | 1, 5 | 1, 6 | 2, 3 | 2, 4 | 2, 5 | 2, 6 |
|---|---|---|---|---|---|---|------|------|------|------|------|------|------|------|------|
| 1 | 0 | 1 | 1 | 1 | 1 | 1 | 1 | 1 | 1 | 1 | 1 | 1 | 1 | 1 | 1 |
| 2 | 0 | 1 | 1 | 1 | 1 | 1 | 1 | 1 | 1 | 1 | 1 | 2 | 2 | 2 | 2 |
| 3 | 1 | 1 | 1 | 1 | 1 | 1 | 1 | 1 | 1 | 1 | 1 | 1 | 1 | 1 | 1 |
| 4 | 1 | 1 | 1 | 1 | 1 | 1 | 2 | 2 | 2 | 2 | 2 | 2 | 2 | 2 | 2 |
| 5 | 1 | 1 | 1 | 1 | 1 | 1 | 1 | 2 | 2 | 2 | 2 | 2 | 2 | 2 | 2 |
| 6 | 1 | 1 | 1 | 1 | 1 | 1 | 0 | 2 | 2 | 2 | 2 | 2 | 2 | 2 | 2 |
| 7 | 1 | 1 | 1 | 1 | 1 | 1 | 1 | 1 | 2 | 2 | 2 | 1 | 2 | 2 | 2 |
| 8 | 1 | 1 | 1 | 1 | 1 | 1 | 2 | 2 | 2 | 2 | 2 | 2 | 2 | 2 | 2 |
| 9 | 1 | 1 | 1 | 1 | 1 | 1 | 1 | 1 | 2 | 2 | 2 | 1 | 2 | 2 | 2 |

| r | 3, 4 | 3, 5 | 3, 6 | 4, 5 | 4, 6 | 5, 6 | 1, 2, 3 | 1, 2, 4 | 1, 2, 5 | 1, 2, 6 | 1, 3, 4 | 1, 3, 5 |
|---|------|------|------|------|------|------|---------|---------|---------|---------|---------|---------|
| 1 | 1 | 1 | 1 | 1 | 1 | 1 | 1 | 1 | 1 | 1 | 1 | 1 |
| 2 | 2 | 2 | 2 | 2 | 2 | 2 | 2 | 2 | 2 | 2 | 2 | 2 |
| 3 | 1 | 1 | 1 | 1 | 1 | 1 | 1 | 1 | 1 | 1 | 1 | 1 |
| 4 | 2 | 2 | 2 | 2 | 2 | 2 | 3 | 3 | 3 | 3 | 3 | 3 |
| 5 | 1 | 1 | 1 | 1 | 1 | 1 | 1 | 1 | 1 | 1 | 2 | 2 |
| 6 | 1 | 1 | 1 | 1 | 1 | 1 | 1 | 1 | 1 | 1 | 2 | 2 |
| 7 | 2 | 2 | 2 | 1 | 1 | 1 | 1 | 2 | 2 | 2 | 2 | 2 |
| 8 | 2 | 2 | 2 | 2 | 2 | 2 | 1 | 3 | 3 | 3 | 3 | 3 |
| 9 | 2 | 2 | 2 | 1 | 1 | 1 | 0 | 2 | 2 | 2 | 2 | 2 |

| r | 1, 3, 6 | 1, 4, 5 | 1, 4, 6 | 1, 5, 6 |
|---|---------|---------|---------|---------|
| 1 | 1 | 1 | 1 | 1 |
| 2 | 2 | 2 | 2 | 2 |
| 3 | 1 | 1 | 1 | 1 |
| 4 | 3 | 3 | 3 | 3 |
| 5 | 2 | 2 | 2 | 2 |
| 6 | 2 | 2 | 2 | 2 |
| 7 | 2 | 2 | 2 | 2 |
| 8 | 3 | 3 | 3 | 3 |
| 9 | 2 | 2 | 2 | 2 |

Table 7: Species in the Florida Bay foodweb with biological classification and assigned clusters. Cluster labels and colors correspond to the clusters shown in Figure 2. For InH-partition, in the first-level clustering, the species Roots is the only singleton while in the second-level clustering, the species Kingfisher, Hawksbill Turtle and Manatee are singletons.

| Species | Biological Classification | Cluster (Ours) | Cluster (Benson's [9]) |
|---|---|---|---|
| Roots | producers (no predator) | Singleton | Singleton |
| $2\mu$m Spherical cya | phytoplankton producers | Green | Singleton |
| Synedococcus | phytoplankton producers | Green | Singleton |
| Oscillatoria | phytoplankton producers | Green | Singleton |
| Small Diatoms ($< 20\mu$m) | phytoplankton producers | Green | Singleton |
| Big Diatoms ($> 20\mu$m) | phytoplankton producers | Green | Singleton |
| Dinoflagellates | phytoplankton producers | Green | Singleton |
| Other Phytoplankton | phytoplankton producers | Green | Singleton |
| Free Bacteria | producers | Green | Green |
| Water Flagellates | producers | Green | Green |
| Water Cilitaes | producers | Green | Green |
| Kingfisher | bird (no predator) | Singleton | Singleton |
| Hawksbill Turtle | reptiles (no predator) | Singleton | Singleton |
| Manatee | mammal (no predator) | Singleton | Singleton |
| Rays | fish | Blue | Singleton |
| Bonefish | fish | Blue | Singleton |
| Lizardfish | fish | Blue | Red |
| Catfish | fish | Blue | Blue |
| Eels | fish | Blue | Red |
| Brotalus | fish | Blue | Blue |
| Needlefish | fish | Blue | Yellow |
| Snook | fish | Blue | Singleton |
| Jacks | fish | Blue | Singleton |
| Pompano | fish | Blue | Singleton |
| Other Snapper | fish | Blue | Singleton |
| Gray Snapper | fish | Blue | Singleton |
| Grunt | fish | Blue | Singleton |
| Porgy | fish | Blue | Singleton |
| Scianids | fish | Blue | Singleton |
| Spotted Seatrout | fish | Blue | Singleton |
| Red Drum | fish | Blue | Singleton |
| Spadefish | fish | Blue | Singleton |
| Flatfish | fish | Blue | Blue |
| Filefish | fish | Blue | Singleton |
| Puffer | fish | Blue | Singleton |
| Other Pelagic fish | fish | Blue | Yellow |
| Small Herons & Egrets | bird | Blue | Singleton |
| Ibis | bird | Blue | Singleton |
| Roseate Spoonbill | bird | Blue | Singleton |
| Herbivorous Ducks | bird | Blue | Singleton |
| Omnivorous Ducks | bird | Blue | Singleton |
| Gruiformes | bird | Blue | Singleton |
| Small Shorebird | bird | Blue | Singleton |
| Gulls & Terns | bird | Blue | Singleton |
| Loggerhead Turtle | reptiles (no predator) | Blue | Singleton |
| Sharks | fish (no predator) | Purple | Singleton |
| Tarpon | fish | Purple | Singleton |
| Grouper | fish (no predator) | Purple | Singleton |
| Mackerel | fish (no predator) | Purple | Singleton |
| Barracuda | fish | Purple | Singleton |
| Loon | bird (no predator) | Purple | Singleton |
| Greeb | bird (no predator) | Purple | Singleton |
| Pelican | bird | Purple | Singleton |
| Comorant | bird | Purple | Singleton |
| Big Herons & Egrets | bird | Purple | Singleton |
| Predatory Ducks | bird (no predator) | Purple | Singleton |
| Raptors | bird (no predator) | Purple | Singleton |
| Crocodiles | reptiles (no predator) | Purple | Singleton |
| SingleDolphin | mammal (no predator) | Purple | Singleton |

| Species | Biological Classification | Cluster Labels | Cluster(Benson's [9]) |
|---|---|---|---|
| Benthic microalgea | algea producers | Yellow | Blue |
| Thalassia | seagrass producers | Yellow | Blue |
| Halodule | seagrass producers | Yellow | Blue |
| Syringodium | seagrass producers | Yellow | Blue |
| Drift Algae | algea producers | Yellow | Blue |
| Epiphytes | algea producers | Yellow | Blue |
| Acartia Tonsa | zooplankton invertebrates | Yellow | Green |
| Oithona nana | zooplankton invertebrates | Yellow | Green |
| Paracalanus | zooplankton invertebrates | Yellow | Green |
| Other Copepoda | zooplankton invertebrates | Yellow | Green |
| Meroplankton | zooplankton invertebrates | Yellow | Green |
| Other Zooplankton | zooplankton invertebrates | Yellow | Green |
| Benthic Flagellates | invertebrates | Yellow | Blue |
| Benthic Ciliates | invertebrates | Yellow | Blue |
| Meiofauna | invertebrates | Yellow | Blue |
| Sponges | macro-invertebrates | Yellow | Green |
| Bivalves | macro-invertebrates | Yellow | Blue |
| Detritivorous Gastropods | macro-invertebrates | Yellow | Blue |
| Epiphytic Gastropods | macro-invertebrates | Yellow | Singleton |
| Predatory Gastropods | macro-invertebrates | Yellow | Blue |
| Detritivorous Polychaetes | macro-invertebrates | Yellow | Blue |
| Predatory Polychaetes | macro-invertebrates | Yellow | Blue |
| Suspension Feeding Polych | macro-invertebrates | Yellow | Blue |
| Macrobenthos | macro-invertebrates | Yellow | Blue |
| Benthic Crustaceans | macro-invertebrates | Yellow | Blue |
| Detritivorous Amphipods | macro-invertebrates | Yellow | Blue |
| Herbivorous Amphipods | macro-invertebrates | Yellow | Blue |
| Isopods | macro-invertebrates | Yellow | Blue |
| Herbivorous Shrimp | macro-invertebrates | Yellow | Red |
| Predatory Shrimp | macro-invertebrates | Yellow | Blue |
| Pink Shrimp | macro-invertebrates | Yellow | Blue |
| Thor Floridanus | macro-invertebrates | Yellow | Singleton |
| Detritivorous Crabs | macro-invertebrates | Yellow | Red |
| Omnivorous Crabs | macro-invertebrates | Yellow | Blue |
| Green Turtle | reptiles | Yellow | Singleton |
| Coral | macro-invertebrates | Red | Singleton |
| Other Cnidaridae | macro-invertebrates | Red | Blue |
| Echinoderma | macro-invertebrates | Red | Blue |
| Lobster | macro-invertebrates | Red | Singleton |
| Predatory Crabs | macro-invertebrates | Red | Red |
| Callinectus sapidus | macro-invertebrates | Red | Red |
| Stone Crab | macro-invertebrates | Red | Singleton |
| Sardines | fish | Red | Yellow |
| Anchovy | fish | Red | Yellow |
| Bay Anchovy | fish | Red | Yellow |
| Toadfish | fish | Red | Blue |
| Halfbeaks | fish | Red | Yellow |
| Other Killifish | fish | Red | Singleton |
| Goldspotted killifish | fish | Red | Yellow |
| Rainwater killifish | fish | Red | Yellow |
| Sailfin Molly | fish | Red | Singleton |
| Silverside | fish | Red | Yellow |
| Other Horsefish | fish | Red | Singleton |
| Gulf Pipefish | fish | Red | Singleton |
| Dwarf Seahorse | fish | Red | Singleton |
| Mojarra | fish | Red | Singleton |
| Pinfish | fish | Red | Singleton |
| Parrotfish | fish | Red | Singleton |
| Mullet | fish | Red | Blue |
| Blennies | fish | Red | Blue |
| Code Goby | fish | Red | Red |
| Clown Goby | fish | Red | Red |
| Other Demersal Fish | fish | Red | Blue |