[Reviews · NeurIPS 2017]

Reviewer 1



This paper considers the hypergraph clustering problem in a more general setting where the cost of hyperedge cut depends on the partitioning of hyperedge (i.e., all cuts of the hyperedge are not treated the same). An algorithm is presented for minimizing the normalized cut in this general setting. The algorithm breaks down for general costs of the hyperedge cut; however the authors derive conditions under which the algorithm succeeds and has provable approximation guarantees. Detailed comments: ================== The main contributions of the paper are + Generalization of hypergraph partitioning to include inhomogeneous cut of the hyper edge; the motivation for this is clearly established. + A novel technique to minimize the normalized cut for this problem. Similar to the existing techniques [e.g., 11], the algorithm tries to derive appropriate pairwise edges so that the problem can be transformed to the standard graph clustering (the approximation guarantees then follow from those of the standard spectral clustering). Few comments on the positives/negative of the paper/algorithm: + The paper is well written and the motivation for the inhomogeneous cut of the hyper edge is clearly established. The algorithm details are easy to follow. - The procedure that finds the approximate graph edge weights does not succeed in all cases as clearly illustrated in the paper. + I liked the part where they derive the conditions under which the procedure is feasible. However, it is not clear how practical it is to enforce such constraints in practice. - The main drawback is that they transform the problem on hypergraphs to graphs and then use the graph algorithms to solve the resulting problem. Although this is one of the standard approaches, it is shown that an exact representation of the hypergraph via a graph retaining its cut properties is impossible (E. Ihler, D. Wagner, and F. Wagner. Modeling hypergraphs by graphs with the same mincut properties. Information Processing Letters, 1993). So a purely hypergraph based algorithm similar to (Hein et. al. The Total Variation on Hypergraphs - Learning on Hypergraphs Revisited. NIPS 2013) would be really interesting in this general setting.

Reviewer 2



In this paper the authors consider a novel version of hypergraph clustering. Essentially, the novelty of the formulation is the following: the cost of a cut hyperedge depends on which nodes go on each side of the cut. This is quantified by a weight function over all possible subsets of nodes for each hyperedge. The complexity of such formulation may at first glance seem unnecessary, but the authors illustrate real-world applications where it makes sense to use their formulation. The paper is nicely written. My main critique is that some key ideas of the paper -as the authors point out themselves- are heavily based on [11]. Also some theoretical contributions are weak, both from a technical perspective and with respect to their generality. For instance, Theorem 3.1 is not hard to prove, and is true under certain conditions that may not hold. On the positive side, the authors do a good work with respect to analyzing the shortcomings of their method.Also, the experimental results are convincing to some extent. My comments follow: 1. The authors provide a motivational example in the introduction. I suggest that the caption under figure 1 should be significantly shortened. The authors may refer the reader to Section 1 for details instead. 2. The authors should discuss in greater details hypergraph clustering, and specifically the formulation where we pay 1 for each edge cut. Ξ€here is no canonical matrix representation of hypergraphs, and optimizing over tensors is NP-hard. The authors should check the paper "Hypergraph Markov Operators, Eigenvalues and Approximation Algorithms" by Anand Louis (arxiv:1408.2425) 3. As pointed out by [21] there is a random walk interpretation for motif-clustering. Is there a natural random walk interpretation of the current formulation in the context of motif clustering? 4. The experimental results are convincing to some extent. It would have been nicer to see more results on motif-clustering. Is there a class of motifs and graphs where your method may significantly outperform existing methods [10,21]? Finally, there is a typo in the title, and couple of others (e.g., "is guaranteed to find a constant-approximation solutions"->"is guaranteed to find a constant-approximation solution") along the way. Use ispell or some other similar program to detect all typos, and correct them.